# Toward Causal-Aware RL:
# State-Wise Action-Refined Temporal Difference

## Abstract

Although it is well known that exploration plays a key role in Reinforcement Learning (RL), prevailing exploration strategies for continuous control tasks in RL are mainly based on naive isotropic Gaussian noise regardless of the causality relationship between action space and the task and consider all dimensions of actions equally important. In this work, we propose to conduct interventions on the primal action space to discover the causal relationship between the action space and the task reward. We propose the method of State-Wise Action Refined (SWAR), which addresses the issue of action space redundancy and promote causality discovery in RL. We formulate causality discovery in RL tasks as a state-dependent action space selection problem and propose two practical algorithms as solutions. The first approach, TD-SWAR, detects task-related actions during temporal difference learning, while the second approach, Dyn-SWAR, reveals important actions through dynamic model prediction. Empirically, both methods provide approaches to understand the decisions made by RL agents and improve learning efficiency in action-redundant tasks.

## 1 Introduction

Although model-free RL has achieved great success in various challenging tasks and outperforms experts in most cases [21, 26, 17, 34, 4], the design of action space always requires elaboration. For example, in the game StarCraftII, hundreds of units can be selected and controlled to perform various actions. To tackle the difficulty in exploration caused by the extremely large action and state space, hierarchical action space design and imitation learning are used [27, 34] to reduce the exploration space. Both of those approaches require expert knowledge of the task. On the other hand, even in the context of imitation learning where expert data is assumed to be accessible, causal confusion will still hinder the performance of an agent [8]. Those defects motivate us to explore the causality-awareness of an agent that permits an agent to discover the causal relationship for the environment and select useful dimensions of action space during policy learning in pursuance of improved learning efficiency. Another motivating example is the in-hand manipulation tasks [2]: robotics equipped with touch sensors outperforms the policies learned without sensors by a clear margin in hand-in manipulation tasks [20], showing the importance of causality discovery between actions and feedbacks in RL. A similar example can be found in human learning: knowing nothing about how to control the finger joints flexibly may not hinder a baby learns to walk, and a baby has not learned how to walk can still learn to use forks and spoons skillfully, inspiring us to believe that the challenge for exploration can be greatly eased after the causality between action space and the given task is learned.

In this work, the recent advance of instance-wise feature selection technique [38] is improved to be more suitable in large-scale state-wise action selection tasks and adapted to the time-series causal discovery setting to select state-conditioned action space in RL with redundant action space. With the

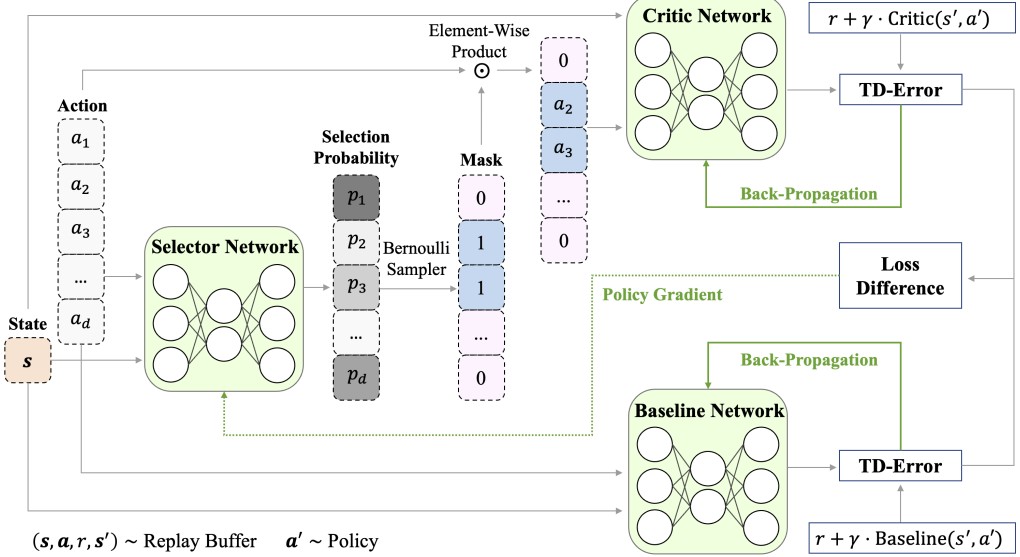

Figure 1: **Block diagram of INVASE in temporal difference learning**. States and actions sampled from replay buffer are fed into the selector network that predicts the selection probabilities of different dimensions of actions. A selection mask is then generated according to such a selection probability vector. The critic network and the baseline network are trained to minimize temporal difference error with states and the selected dimension of actions and primal action respectively. The difference of TD-Error is used to conduct a policy gradient to update the selector network.

proposed method, the agent learns to perform intervention, discover the true structural causal model (SCM) and select task-related actions for a given task, remarkably reduces the burden of exploration and obtains on-par learning efficiency as well as asymptotic performance compared with agents trained in the oracle settings where the action spaces are pruned according to given tasks manually.

## 2    Preliminary

**Markov Decision Processes**   RL tasks can be formally defined as Markov Decision Processes (MDPs), where an agent interacts with the environment and learns to make decision at every timestep. Formally, we consider the deterministic MDP with a fixed horizon $H \in \mathbb{N}^+$ denoted by a tuple $(\mathcal{S}, \mathcal{A}, H, r, \gamma, \mathcal{T}, \rho_0)$, where $\mathcal{S}$ and $\mathcal{A}$ are the $|\mathcal{S}|$-dimensional state and $|\mathcal{A}|$-dimensional action space; $r : \mathcal{S} \times \mathcal{A} \mapsto \mathbb{R}$ denotes the reward function; $\gamma \in (0, 1]$ is the discount factor indicating importance of present returns compared with long-term returns; $\mathcal{T} : \mathcal{S} \times \mathcal{A} \mapsto \mathcal{S}$ denotes the transition dynamics; $\rho_0$ is the initial state distribution.

We use $\Pi$ to represent the stationary deterministic policy class, i.e., $\Pi = \{\pi : \mathcal{S} \mapsto \mathcal{A}\}$. The learning objective of an RL algorithm is to find $\pi^* \in \Pi$ as the solution of the following optimization problem: $\max_{\pi \in \Pi} \mathbb{E}_{\tau \sim \rho_0, \pi, \mathcal{T}}[\sum_{t=1}^{H} \gamma^t r_t]$ where the expectation is taken over the trajectory $\tau = (s_1, a_1, r_1, \ldots, s_H, a_H, r_H)$ generated by policy $\pi$ under the environment $\mathcal{T}$, starting from $s_0 \sim \rho_0$.

**INVASE**   INVASE is proposed by [38] to perform instance-wise feature selection to reduce over-fitting in predictive models. The learning objective is to minimize the KL-Divergence of the full-conditional distribution and the minimal-selected-features-only conditional distribution of the outcome, i.e., $\min_F \mathcal{L}$, with

$$\mathcal{L} = \mathcal{D}_{KL}(p(Y|X = x)||p(Y|X^{(F(x))} = x^{(F(x))}))) + \lambda|F(x)|_0. \tag{1}$$

where $F : \mathcal{X} \to \{0, 1\}^d$ is a feature selection function and $|F(x)|_0$ denotes the cardinality ($l_0$ norm) of selected features, i.e., the number of 1's in $F(x)$. [1] $d$ is the dimension of input features.

---

[1]To avoid confusion between state notion $s \in \mathcal{S}$ and the selector notion $S$ used in [38], $F$ is used in this work to represent the selector (i.e., mask generator).

$x^{(F(x))} = F(x) \odot x$ denotes the element-wise product of $x$ and generated mask $m = F(x)$. Ideally, the optimal selection function $F$ should be able to minimize the two terms in Equation (1) simultaneously.

INVASE applies the Actor-Critic framework in the optimization of $F$ through sampling, where $f_\theta(\cdot|x)$, parameterized by a neural network $\theta$ [2], is used as a stochastic actor. Two predictive networks $C_\phi(\cdot), B_\psi(\cdot)$ are considered as the critic and the baseline network used for variance reduction [36] and trained with the Cross-Entropy loss to produce return signal $\mathcal{L}$, based on which $f_\theta(\cdot|x)$ can be optimized through policy gradient:

$$\mathbb{E}_{(x,y)\sim p}[\mathbb{E}_{m\sim f_\theta(\cdot|x)}[\mathcal{L}\nabla_\theta \log f_\theta(\cdot|x)]]. \tag{2}$$

Finally, $F(x) = (F_1(x), ..., F_d(x))$ can be get by sampling from $f(\cdot|x) = (f_1(\cdot|x), ..., f_d(\cdot|x))$, with

$$F_i(x) = \begin{cases} 1, & \textbf{w.p.} & f_i(\cdot|x). \\ 0, & \textbf{w.p.} & 1 - f_i(\cdot|x). \end{cases} \tag{3}$$

# 3 Proposed Method

The objective of this work is to carry out state-wise action selection in RL through intervention, and thereby enhance the learning efficiency with a pruned task-related action space after finding the correct causal model. Section 3.1 starts with the formalization of the action space refinery objective in RL tasks under the framework of causal discovery. Section 3.2 introduces SWAR, which improves the scalability of INVASE in high dimensional variable selection tasks. We integrate SWAR with deterministic policy gradient methods [25] in Section 3.3 to perform state-wise action space pruning, resulting in two practical causality-aware RL algorithms.

## 3.1 Temporal Difference Objective with Structural Causal Models

In modern RL algorithms, the most general approach is based on the Actor-Critic framework [15], where the critic $Q_w(s, a)$ approximates the return of given state-action pair $(s, a)$ and guides the Actor to maximize the approximated return at state $s$. The Critic is optimized to reduce Temporal Difference (TD) error [29], defined as

$$\mathcal{L}_{TD} = \mathbb{E}_{s_i, a_i, r_i, s'_i \sim \mathcal{B}}[(r_i + \gamma Q_w(s'_i, a'_i) - Q_w(s_i, a_i))^2]. \tag{4}$$

where $\mathcal{B} = (s_i, a_i, r_i, s'_i)_{i=1,2,...}$ is the replay buffer used for off-policy learning [17, 10, 12, 28], and $a'_i = \pi(s'_i)$ is the predicted action for state $s'_i$. In practice, the calculations of $Q_w(s'_i, a'_i)$ are usually based on another set of slowly updated target networks for stability [10, 12]. Henceforth, TD-learning can be roughly simplified as regression with notion $y_i = r_i + \gamma Q_w(s'_i, a'_i)$:

$$\mathcal{L}_{TD} = \mathbb{E}_{s_i, a_i, r_i, s'_i \sim \mathcal{B}}[(y_i - Q_w(s_i, a_i))^2]. \tag{5}$$

Assume there are only $M < L$ actions are related to a specific task among the $L$-dimensional actions $a_i = a_i^{(1)}, ..., a_i^{(L)}$, i.e., $Q_w(\cdot, \cdot)$ is function of $s_i, a_i^{(1)}, ..., a_i^{(M)}$. Learning with the primal redundant action space will lead to around $\frac{L+|\mathcal{S}|}{M+|\mathcal{S}|}$ times sample complexity [9, 39]. Therefore, we are motivated to improve the learning efficiency of $Q$ by pruning those task-irrelevant action dimensions $a_i^{(M+1)}, ..., a_i^{(L)}$ by finding an action selection function $G$, satisfying

$$\min_{G,Q_w} \mathbb{E}_{s_i, a_i, r_i, s'_i \sim \mathcal{B}}[(y'_i - Q_w(s_i, a_i^{(G(a_i|s_i))}))^2] + \lambda|G(a_i|s_i)|_0. \tag{6}$$

where $y'_i = r_i + \gamma Q_w(s'_i, a_i^{'(G(a'_i|s_i))})$.

Such a problem can be addressed from the perspective of causal discovery. Formally, we can use the Structural Causal Models (SCMs) to represent the underlying causal structure of a sequential decision making process, as shown in Figure 2. Under this language, we use the notion of **causal** actions to denote $a_i^{(1,...,M)}$, and **nuisance** actions for other dimension of actions. In our work, we use IC-INVASE for causal discovery. Ideally, the action selection function $G$ should be able to distinguish between nuisance action dimensions and the causal ones that has causal relation with either dynamics or reward mechanism. We present in the next section our causal discovery algorithms.

---

[2]In this work, subscripts $\phi, \psi, \theta, w$ are used to denote the parameter of neural networks.

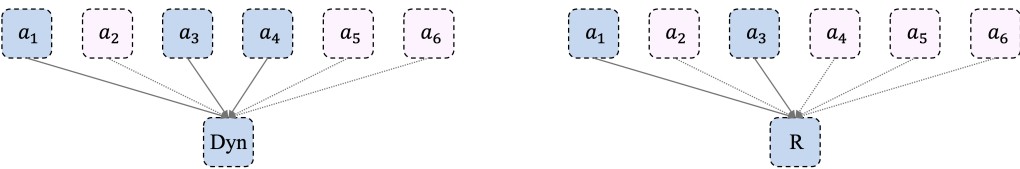

Figure 2: SCM of temporal difference learning. Among all executable actions, there can be only a subset have effect on the dynamical changes or the reward mechanism. In our work, we use IC-INVASE as a causal discovery tool to distinguish the causal irrelevant actions and hence improve learning efficiency.

### 3.2 Iterative Curriculum INVASE (IC-INVASE)

Instead of directly applying INVASE to solve Equation (6). We first propose two improvements to make the vanilla INVASE more suitable for large-scale variable selection tasks as the action dimension in RL might be extremely large [34]. Specifically, the first improvement, based on curriculum learning, is introduced to tackle the exploration difficulty when $\lambda$ in Equation (1) is large, where INVASE tends to converge to poor sub-optimal solutions and prune all variables including the useful ones [38]. The second improvement is based on the iterative structure of variable selection tasks: the feature selection operator $G$ can be applied multiple times to conduct hierarchical feature selection without introducing extra computation expenses.

#### 3.2.1 Curriculum Learning For High Dimensional Variable Selection

The work of [3] first introduces Curriculum Learning to mimic human learning by gradually learn more complex concepts or handle more difficult tasks. Effectiveness of the method has been demonstrated in various set-ups [3, 19, 7, 35, 37]. In general, it should be easier to select $M$ useful variables out of $L$ input variables when $M$ is larger. The most trivial case is to select all $L$ variables, with an identical mapping $x^{(G(x))} = G(x) \odot x = x$. Formally, we have

**Proposition 1** (Curriculum Property in Variable Selection)**.** *Assume $M$ out of $L$ variables are outcome-related, let $M \leq N_1 < N_2 \leq L$, $G_{N_1}(x)$ minimizes $\mathcal{D}_{KL}(p(Y|X = x)||p(Y|X^{(G(x))} = x^{(G(x))})) + \lambda||G(x)|_0 - N_1|$. Then $G_{N_2}(x)$ minimizes $\mathcal{D}_{KL}(p(Y|X = x)||p(Y|X^{G(x)} = x^{G(x)})) + \lambda||G(x)|_0 - N_2|$ can be get through: $G_{N_2}(x) \in \{G_{N_1}(x) \vee [G_{N_1}(x)\textbf{XOR 1}]_{1_{N_2 - N_1}}\}$, where $[\cdot]_{1_{N_2 - N_1}}$ means keep $N_2 - N_1$ none-zero elements unchanged while replacing other elements by $0$.*

*Proof.* By the definition of the $[\cdot]_{1_{N_2 - N_1}}$ operator, $||G(x)|_0 - N_2| = 0$ is minimized. On the other hand, starting from $N_1 = M$, minimizing $\mathcal{D}_{KL}(p(Y|X = x)||p(Y|X^{(G(x))} = x^{(G(x))}))$ requires all the $M$ outcome-related variables being selected by $G_{N_1}$. Therefore, $G_{N_2}$ also minimizes the KL-divergence by the independent assumption of the other $L - M$ variables with the outcomes. $\square$

The proposition indicates the difficulty of selecting $N$ useful out of $L$ variables decreases monotonically as $N \geq M$ increase from $M, M + 1, ..., L$. In this work, two classes of practical curriculum are designed: 1. curriculum on the $l_0$ penalty coefficient, and 2. curriculum on the proportion of variables to be selected.

**Curriculum on $l_0$ Penalty Coefficient** In this curriculum design, the penalty coefficient $\lambda$ in Equation (1) is increased from $0$ to a pre-determined number (e.g., $1.0$). Increasing the value of $\lambda$ will lead to a larger penalty on the number of variables selected by the feature selector. Experiments in [38] has shown a large $\lambda$ always lead to a trivial selector that does not select any variable.

**Curriculum on the Proportion of Selected Features** In this curriculum design, the proportion of variables to be selected, denoted by $p_r$, is adjusted from the default setting $0$ to a decreasing number

---

**Algorithm 1** TD3 with TD-SWAR

---

Initialize critic networks $C_{\phi_1}, C_{\phi_2}$, baseline networks $B_{\psi_1}, B_{\psi_2}$ and actor network $\pi_\nu$, IC-INVASE
selector network $G_\theta$
Initialize target networks $\phi_1' \leftarrow \phi_1, \phi_2' \leftarrow \phi_2, \psi_1' \leftarrow \psi_1, \psi_2' \leftarrow \psi_2, \nu' \leftarrow \nu$
Initialize replay buffer $\mathcal{B}$
**for** $t = 1, H$ **do**
    Interact with environment and store transition tuple $(s, a, r, s')$ in $\mathcal{B}$
    Sample mini-batch of transitions $\{(s, a, r, s')\}$ from $\mathcal{B}$
    Calculate perturbed next action by $\tilde{a} \leftarrow \pi_{\nu'}(s') + \epsilon$, $\epsilon$ is sampled from a clipped Gaussian.
    Select actions with target selector network
    $\tilde{a}^{(G(\tilde{a}|s'))} \leftarrow G_{\theta'}(\tilde{a}|s') \odot \tilde{a}$
    Calculate target critic value $y_c$ and baseline value $y_b$:
    $y_c \leftarrow r + \gamma \min_{i=1,2} C_{\phi_i'}(s', \tilde{a}^{(G(\tilde{a}|s'))})$
    $y_b \leftarrow r + \gamma \min_{i=1,2} B_{\psi_i'}(s', \tilde{a})$
    Update critics and baselines with selected actions:
    $a^{(G(a|s))} \leftarrow G_\theta(a|s') \odot a$
    $\phi_i \leftarrow \arg\min_{\phi_i} \mathbf{MSE}(y_c, C_{\phi_i}(s, a^{(G(a|s))}))$
    $\psi_i \leftarrow \arg\min_{\psi_i} \mathbf{MSE}(y_b, B_{\psi_i}(s, a))$
    Update IC-INVASE selector network by the policy gradient, with learning rate $\eta_1$:
    $\theta \leftarrow \theta - \eta_1(l_b - l_c)\nabla_\theta \log G_\theta(a|s)$, $l_b, l_c$ are MSE
    losses in the previous step.
    Update $\nu$ by the deterministic policy gradient, with learning rate $\eta_2$:
    $\nu \leftarrow \nu - \eta_2 \nabla_a C_{\phi_1}(s, a)|_{a=\pi_\nu(s)} \nabla_\nu \pi_\nu(s)$
    Update target networks, with $\tau \in (0, 1)$:
    $\phi_i' \leftarrow \tau\phi_i + (1-\tau)\phi_i'$
    $\psi_i' \leftarrow \tau\psi_i + (1-\tau)\psi_i'$
    $\nu' \leftarrow \tau\nu + (1-\tau)\nu'$
**end for**

---

from a pre-determined value (e.g., 0.5) to 0. i.e., the $l_0$ penalty term $\lambda|G(x)|_0$ in Equation (1) is revised to be $\lambda||G(x)|_0 - d \cdot p_r|$, where $d$ is the dimension of input $x$. When the proportion is set to be $p_r = 0.5$, the selector will be penalized whenever less or more than half of all variables are selected. Such a curriculum design forces the feature selector to learn to select less but increasingly more important variables gradually.

Thus, we get the learning objective of curriculum-INVASE:

$$\mathcal{L} = \mathcal{D}_{KL}(p(Y|X = x)||p(Y|X^{(G(x))} = x^{(G(x))}))) + \lambda||G(x)|_0 - d \cdot p_r|. \tag{7}$$

where $\lambda$ increases from 0 to some value and $p_r$ decreases from a value in $[0, 1)$ to 0.

### 3.2.2 Iterative Variable Selection

The second improvement proposed in this work is based on the iterative structure of variable selection tasks. Specifically, the $G(x)$ mapping $x \in \mathcal{X}$ to $\{0, 1\}^d$ is an iterative operator, which can be applied for multiple times to perform coarse-to-fine variable selection. Although in practice we follow [38] to apply an element-wise product in producing $x^{(G(x))}$: $x^{(G(x))} = G(x) \odot x \in \mathcal{X}$. In more general cases, the i-th element of $x_i^{(G(x))}$ is

$$x_i^{(G(x))} = \begin{cases} 1, & \text{if} \quad G_i(x) = 1. \\ *, & \text{if} \quad G_i(x) = 0. \end{cases} \tag{8}$$

where $*$ can be an arbitrary identifiable indicator that represents the variable is not selected.

On the other hand, once the outputs $G(x)$ of the selector have been recorded, $*$ can be replaced by any label-independent variable $G(x) \odot z$, where $z \sim p_z(\cdot)$ is outcome-independent. Then $x^{(G(x))}$ can be regarded as a new sample and be fed into the variable selector, resulting in a hierarchical

variable selection process:

$$
\begin{aligned}
x^{(1)} &= (G(x) \odot x) \oplus (G(x) \odot z), \\
x^{(2)} &= (G(x^{(1)}) \odot x^{(1)}) \oplus (G(x^{(1)}) \odot z), \\
&\cdots \\
x^{(n)} &= (G(x^{(n-1)}) \odot x^{(n-1)}) \oplus (G(x^{(n-1)}) \odot z),
\end{aligned}
\tag{9}
$$

where $z \sim p_z(\cdot)$, and $\oplus$ is the element-wise sum operator. Moreover, if the distribution of irrelevant variable $p_x(\cdot)$ is known, applying the variable selection operator obtained from Equation (7) for multiple times with $p_z(\cdot) \overset{d}{=} p_x(\cdot)$ has the meaning of hierarchical variable selection: after each operation, the most obvious $1 - p_r$ irrelevant variables are discarded. e.g., when $p_r = 0.5$, ideally top-$50\%, 25\%, 12.5\%$ most important variables will be selected after the first three selection operations. In this work, a coarse approximation is utilized by selecting $z$ to be $z = 0$ for simplicity. [3]

Combining those two improvements lead to an Iterative Curriculum version of INVASE (IC-INVASE) that addresses the exploration difficulty in high-dimensional variable selection tasks. Curriculum learning helps IC-INVASE to achieve better asymptotic performance, i.e., achieve higher True Positive Rate (TPR) and lower False Discovery Rate (FDR), while iterative application of the selection operator contributes to higher learning efficiency: selectors models with different level of TPR/FDR can be generated on-the-fly.

### 3.3  State-Wise Action Refinery with IC-INVASE

#### 3.3.1  Temporal Difference State-Wise Action Refinery

With the techniques introduced in the previous section, higher dimensional variable selection tasks can be better solved, therefore we are ready to use IC-INVASE to solve Equation (6). The resulting algorithm is called Temporal Difference State-Wise Action Refinery (TD-SWAR).

In this work, TD3 [10] is used as the basic algorithm we build TD-SWAR up on. In addition to the policy network $\pi_\nu$, double critic networks $C_{\phi_1}, C_{\phi_2}$ and their corresponding target networks used in vanilla TD3, TD-SWAR includes an action selector model $G_\theta$ and two baseline networks $B_{\psi_1}$, $B_{\psi_2}$ following [38] to reduce the variance in policy gradient learning. Pseudo-code for the proposed algorithm is shown in Algorithm 1. And the block diagram in Figure 1 illustrates how different modules in TD-SWAR updates their parameters.

#### 3.3.2  Static Approximation: Model-Based Action Selection

While IC-INVASE can be formally integrated with temporal difference learning, the learning stability is not guaranteed. Different from general regression tasks where the label for every instance is fixed across training, in temporal difference learning, the regression target is closely related to the present critic function $C_\phi$, the policy $\pi_\nu$ that generates the transition tuples used for training, and the selector model of IC-INVASE itself. In this section, a static approach is proposed to approximately solve the challenge of instability in TD-SWAR [4].

Other than applying the IC-INVASE algorithm to solve Equation (6), another way of leveraging IC-INVASE in action space pruning is to combine it with the model-based methods [11, 16, 13, 14], where a dynamic model $\mathcal{P} : \mathcal{S} \times \mathcal{A} \mapsto \mathcal{S}$ is learned through regression:

$$
\mathcal{P} = \arg\min_{\mathcal{P}} \mathbb{E}_{(s,a,s') \sim \pi, \mathcal{T}} (s' - \mathcal{P}(s, a))^2
\tag{10}
$$

Although the task of precise model-based prediction is in general challenging [24], in this work, we only adopt model-based prediction in action selection, and the target is action discovery other than precise prediction. As the dynamic models are always static across learning, such an approach can be much more stable than TD-SWAR. We name this method as Dyn-SWAR and present the pseudo-code in Algorithm 2, where we infuse IC-INVASE to Equation (10) and get the learning objective:

$$
\min_{G, \mathcal{P}} \mathbb{E}_{(s,a,s') \sim \pi, \mathcal{T}} (s' - \mathcal{P}(s, a^{(G(a|s))}))^2
\tag{11}
$$

---

[3]$p_z(\cdot)$ may be learned through generative models to approximate $p_x(\cdot)$, and Equation (9) can be regarded as a kind of data-augmentation or ensemble method. This idea is left for the future work.

[4]Analysis on the approximation is provided in Appendix A.

**Algorithm 2** TD3 with Dyn-SWAR

Initialize critic networks $Q_{w_1}, Q_{w_2}$, Dynamics critic model $C_\phi$, dynamic baseline model $B_\psi$, actor network $\pi_\nu$, and IC-INVASE selector network $G_\theta$

Initialize target networks $w_1' \leftarrow w_1, w_2' \leftarrow w_2, \nu' \leftarrow \nu$

Initialize replay buffer $\mathcal{B}$

**for** $t = 1, H$ **do**

    Interact with environment and store transition tuple $(s, a, r, s')$ in $\mathcal{B}$

    Sample mini-batch of transitions $\{(s, a, r, s')\}$ from $\mathcal{B}$

    Update dynamic critics and dynamic baselines with equation (10):

     $\phi \leftarrow \arg\min_\phi \mathbf{MSE}(s', C_\phi(s, a^{(G(a|s))}))$

     $\psi \leftarrow \arg\min_\psi \mathbf{MSE}(s', B_\psi(s, a))$

    Update IC-INVASE selector network by the policy gradient, with learning rate $\eta_1$:

     $\theta \leftarrow \theta - \eta_1(l_b - l_c)\nabla_\theta \log G_\theta(a|s)$, $l_b, l_c$ are MSE

     losses in the previous step.

    Calculate perturbed next action by $\tilde{a} \leftarrow \pi_{\nu'}(s') + \epsilon$, $\epsilon$ is sampled from a clipped Gaussian.

    Select actions with selector network

     $\tilde{a}^{(G(\tilde{a}|s'))} \leftarrow G_{\theta'}(\tilde{a}|s') \odot \tilde{a}$

    Calculate target critic value $y$ and update critic networks:

     $y \leftarrow r + \gamma \min_{i=1,2} Q_{w_i'}(s', \tilde{a}^{(G(\tilde{a}|s'))})$

     $w_i \leftarrow \arg\min_{w_i} \mathbf{MSE}(y, Q_{w_i}(s, a^{(G(a|s))}))$

    Update $\nu$ by the deterministic policy gradient, with learning rate $\eta_2$:

     $\nu \leftarrow \nu - \eta_2 \nabla_a Q_{w_1}(s, a)|_{a=\pi_\nu(s)} \nabla_\nu \pi_\nu(s)$

    Update target networks, with $\tau \in (0, 1)$:

     $w_i' \leftarrow \tau w_i + (1 - \tau)w_i'$

     $\nu' \leftarrow \tau\nu + (1 - \tau)\nu'$

**end for**

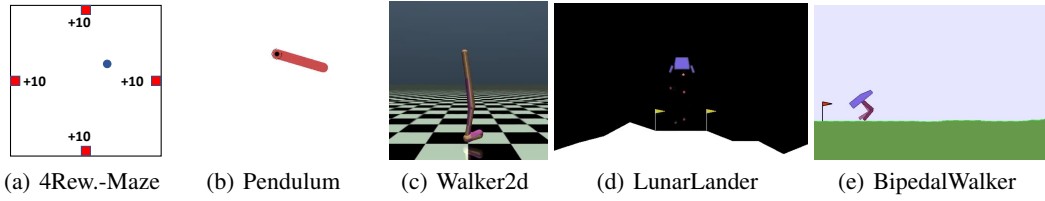

(a) 4Rew.-Maze     (b) Pendulum     (c) Walker2d     (d) LunarLander     (e) BipedalWalker

Figure 3: Environments used in experiments

## 4 Experiment

In this section, we apply our proposed methodologies to five continuous control RL tasks characterized by redundant action spaces, wherein our methods facilitate causality-aware RL. We also present a quantitative comparison between IC-INVASE and the standard INVASE on synthetic datasets in Appendix B, which serves to underscore the enhanced scalability of our approach.

In the present set of experiments, we employed five RL environments (Figure 5), detailed in Table 1 [5]. The symbol $|\mathcal{S}|$ designates the dimension of the state space for each task, while $|\mathcal{A}|$ signifies the dimension of the action space relevant to the task, and $|\mathcal{A}_{red.}|$ represents the dimension of the redundant action space incorporated into each task. These surplus dimensions of actions don't impact state transitions or reward calculations, but it is essential for an agent to identify these redundant dimensions for efficient learning.

We assessed both TD-SWAR, which combines IC-INVASE with temporal difference learning, and its static counterpart, Dyn-SWAR, which employs IC-INVASE in dynamics prediction. The results are benchmarked against two base conditions: the **Oracle**, where redundant action dimensions are

---

[5]For comprehensive descriptions of the environments, please consult Appendix C

Table 1: Tasks used in evaluating SWAR in temporal difference learning

| TASK/DIMENSION | $|\mathcal{S}|$ | $|\mathcal{A}|$ | $|\mathcal{A}_{red.}|$ |
|---|---|---|---|
| PENDULUM-V0 | 3 | 1 | 100 |
| FOURREWARDMAZE | 2 | 2 | 100 |
| LUNARLANDERCONTINUOUS-V2 | 8 | 2 | 100 |
| BIPEDALWALKER-V3 | 24 | 4 | 100 |
| WALKER2D-V2 | 17 | 6 | 100 |

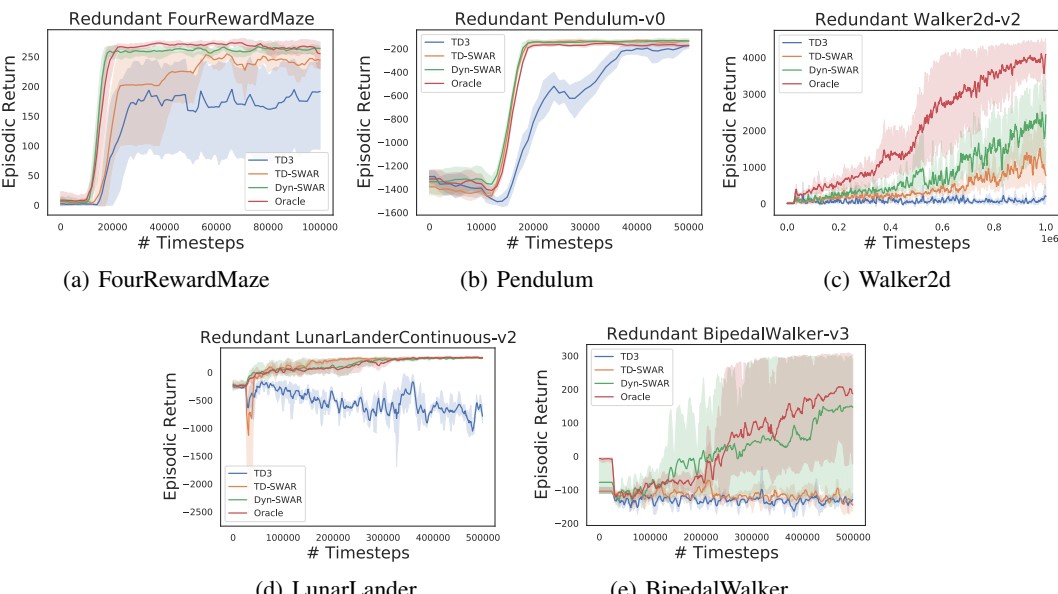

(a) FourRewardMaze

(b) Pendulum

(c) Walker2d

(d) LunarLander

(e) BipedalWalker

Figure 4: Performance of agents in five different environments. The curves shows averaged learning progress and the shaded areas show standard deviation.

manually removed; and **TD3**, which is the standard TD3 algorithm devoid of any explicit action redundancy reduction.

In our experimental findings, we observed that Dyn-SWAR's deployment demonstrates superior efficiency with respect to both sample complexity and computational cost. In contrast, TD-SWAR requires a persistent update of all parameters for the IC-INVASE selector to maintain congruence with the real-time policy and value networks, given the fluctuating regression label over time. However, the Dyn-SWAR selector necessitates a significantly reduced data set for training, specifically between 10,000 and 25,000 timesteps of environmental interaction. This attribute can seamlessly integrate with the warm-up technique utilized in TD3 [10]. Namely, the Dyn-SWAR selector could be trained with warm-up transition tuples gathered during the random exploration phase, and then remain static throughout the subsequent learning process. Compared to traditional RL configurations that generally require millions of environmental interactions, the training of Dyn-SWAR incurs only a minuscule computational cost.

These findings are illustrated in Figure 4. Across all environments, agent learning with IC-INVASE in both TD- and Dyn- methods exceeds the performance of the standard TD3 baseline. Dyn-SWAR achieves a learning efficiency that is on par with oracle benchmarks. However, the performance of TD-SWAR in tasks of higher dimensions (Walker2d-v2 and BipedalWalker-v3) indicates significant potential for enhancement. Accordingly, future work should prioritize enhancing the stability and scalability of instance-wise variable selection within temporal difference learning.

# 5 Related Work

**Instance-Wise Feature Selection** While traditional feature selection method like LASSO [31] aims at finding globally important features across the whole dataset, instance-wise feature selection try to discover the feature-label dependency on a case-by-case basis. L2X [5] performs instance-wise feature selection through mutual information maximization with the technique of Gumbel softmax. L2X requires pre-determined hyper-parameter $k$ to indicate how many features should be selected for each instance, which limits its performance while the number of label-relevant features varies across instances. In this work, we build our instance-wise action selection model on top of INVASE [38], where policy gradient is applied to replace the Gumbel softmax trick and the size of chosen features per instance is more flexible. [32] considers instance-wise feature selection problems in time-series setting, and build generative models to capture counterfactual effects in time series data. Their work enables evaluation of the importance of features over time, which is crucial in the context of healthcare. [18] formally defines different types of feature redundancy and leverages mutual information maximization in instance-wise feature group discovery and introduces theoretical guidance to find the optimal number of different groups.

Our work is distinguished from previous works for instance-wise feature selection in two aspects. First, while previous works focus on static scenarios like classification and regression, this work focus on temporal difference learning where there is no static label. Second, the scalability of previous methods in variable selection is challenged as there might exist hundreds of redundant actions in the context of RL.

**Dimension Reduction in RL** In the context of RL, attention models [33] have been applied to interpret the behaviors of learned policies. [30] proposes to perceive the state information through a self-attention bottleneck in vision-based RL tasks, which concentrates on the state space redundancy reduction with image inputs. The work of [22] also applies the attention mechanism to learn task-relevant information. The proposed method achieves state-of-the-art performance on Atari games with image input while being more understandable with top-down attention models.

Different from those papers, this work considers relatively tight state representations (vector input), and focuses on the task-irrelevant action reduction. We aim at finding the task-related actions and improving the learning efficiency without wasting samples in learning the task-irrelevant dimensions of actions. Our work is most closely related to AE-DQN [39] in that we both consider the problem of redundant action elimination. AE-DQN tackles action space redundancy with an action-elimination network that eliminates sub-optimal actions. Yet its discussion is limited in the discrete settings. In contrast, our work focuses on action elimination in continuous control tasks.

# 6 Conclusion and Future Work

In this study, we address the issue of pruning the action space in action redundant RL tasks. We employ the recent advancements in instance-wise feature selection technology (INVASE), incorporating both curriculum learning and iterative processes, to aim for improved scalability and efficiency. This leads to the creation of the IC-INVASE method, which is then adapted to the RL environment where we introduce two novel algorithms, TD-SWAR and Dyn-SWAR, to implement causality-conscious RL. The former algorithm directly addresses the issue of action redundancy in temporal difference learning, whereas the latter algorithm leverages model-based prediction to capture dynamic causality. Experimental evidence from a range of tasks underscores the importance of causality-awareness for RL agents to achieve efficient learning in action-redundant settings.

As for future research, the iterative characteristic of this method could be further investigated to apply ensemble methods in variable selection. Additionally, the design of a more appropriate curriculum could enhance the fusion of multiple curricula. From the RL perspective, the stability of TD-SWAR could be further optimized to enhance sample efficiency. The design of the curriculum could potentially offer benefits. For instance, an agent might initially learn to identify actions of general importance before concentrating on discerning state-dependent crucial actions. Furthermore, the selection process can be extended to include both the state space and action space, allowing for efficient temporal difference learning that is mindful of the causal relationships among states, actions, and the task at hand. Additionally, model-based prediction could be broadened to anticipate future returns.

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
