# A    On the Dynamic Model Approximation

We provide analysis on the approximation in this section based on the deterministic MDP model in finite action space where the problem degenerates to $Q$-Learning. Similar results can be get to prove the Policy Evaluation Lemma, combined with Policy Improvement Lemma (given proper function approximation of the $\arg\max$ operator) and result in Policy Iteration Theorem.

In deterministic MDPs with $s_{t+1} = \mathcal{T}(s_t, a_t)$, $r_t = r(s_t, a_t)$, the value function of a state is defined as

$$V^\pi(s) = \sum_{t=0}^{\infty} \gamma^t r(s_t, a_t), \tag{12}$$

given $s_0 = s$ is the initial state and $a_t = \pi(s_t)$ comes from the deterministic policy $\pi$.

The learning objective is to find an optimal policy $\pi$, such that an optimal state value can be achieved:

$$V^*(s) = \max_\pi V^\pi(s) \tag{13}$$

The state-action value function ($Q$-function) is then defined as

$$Q(s, a) = r(s, a) + \gamma V^*(\mathcal{T}(s, a)) \tag{14}$$

Formally, the objective of action space pruning in action-redundant MDPs is to find an optimal policy $\pi^{(G)} = G(\pi(s_t)|s_t) \odot \pi(s_t)$ with an action selector $G : \mathcal{S} \times \mathcal{A} \mapsto \{0, 1\}^d$,

$$V^*(s) = \max_{\pi^{(G)}} V^{\pi^{(G)}}(s) = \max_\pi V^\pi(s), \tag{15}$$

with minimal number of actions selected, i.e., $|G|_0$ is minimized. The sufficient and necessary condition for Equation (15) to hold is $r(s_t, \pi(s_t)) = r(s_t, \pi^{(G)}(s_t))$ and $\mathcal{T}(s_t, \pi(s_t)) = \mathcal{T}(s_t, \pi^{(G)}(s_t))$.

In general, the reward function $r$ and transition dynamics $\mathcal{T}$ may depend on different subsets of actions and the optimal, i.e., $r(s_t, a_t) = r(s_t, a_t^{(G_1)})$, while $\mathcal{T}(s_t, a_t) = \mathcal{T}(s_t, a_t^{(G_2)})$, where $G_1, G_2$ select different subset of given actions by $a_t^{(G_1)} = G_1(a_t|s_t) \odot a_t$, $a_t^{(G_2)} = G_2(a_t|s_t) \odot a_t$ but $a_t^{(G_1)} \neq a_t^{(G_2)}$. The final action selector $G$ should be generated according to $G(a|s) = G_1(a|s) \vee G_2(a|s)$, where $\vee$ is the element-wise **OR** operation.

Therefore, in our approximation of Dyn-SWAR, we assume $G(a|s) = G_2(a|s)$ as an approximation for $G(a|s) = G_1(a|s) \vee G_2(a|s)$. Future work may include another predictive model for the reward function and take the element-wise **OR** operation to get $G$.

# B    Additional Experiments

## B.1    Synthetic Data Experiment

The synthetic datasets are generated in the same way as [5, 38]. Specifically, there are 6 synthetic datasets that have inputs generated from an 11-dim Gaussian distribution without correlations across features. The label $Y$ for each dataset is generated by a Bernoulli random variable with $P(Y = 1|X) = \frac{1}{1+\text{logit}(X)}$. In different tasks, $\text{logit}(X)$ takes the value of:

- **Syn1**: $\exp(X_1 X_2)$
- **Syn2**: $\exp(\sum_{i=3}^{6} X_i^2 - 4)$
- **Syn3**: $-10 \times \sin 2X_7 + 2|X_8| + X_9 + \exp(-X_{10})$
- **Syn4**: if $X_{11} < 0$, logit follows **Syn1**, otherwise, logit follows **Syn2**
- **Syn5**: if $X_{11} < 0$, logit follows **Syn1**, otherwise, logit follows **Syn3**
- **Syn6**: if $X_{11} < 0$, logit follows **Syn2**, otherwise, logit follows **Syn3**

In the first three synthetic datasets, the label $Y$ depends on the same feature across each dataset, while in the last three datasets, the subsets of features that label $Y$ depends on are determined by the values of $X_{11}$.

Table 2: Relevant variables discovery results for Synthetic datasets with 11-dim input

| DATA SET | METHOD | ITERATION 1 | | ITERATION 2 | | ITERATION 3 | | ITERATION 4 | |
|---|---|---|---|---|---|---|---|---|---|
| METRIC | | TPR | FDR | TPR | FDR | TPR | FDR | TPR | FDR |
| **Syn4** | INVASE (REP.) | 99.8 | 10.3 | | | | | | |
| | INVASE (EXP.) | 98.6 | 1.6 | 98.1 | 1.1 | 98.1 | 1.1 | 98.1 | 1.1 |
| | IC-INVASE ($\lambda \uparrow 0.2$) | 99.7 | 3.4 | 99.7 | 2.6 | 99.7 | 2.5 | 99.7 | 2.5 |
| | IC-INVASE ($\lambda \uparrow 0.3$) | 99.3 | 1.6 | **99.3** | **0.8** | **99.3** | **0.8** | **99.3** | **0.8** |
| **Syn5** | INVASE (REP.) | 84.8 | 1.1 | | | | | | |
| | INVASE (EXP.) | 82.1 | 1.0 | 79.7 | 1.0 | 79.3 | 1.0 | 79.2 | 1.0 |
| | IC-INVASE ($\lambda \uparrow 0.2$) | 99.3 | 1.6 | 99.1 | 1.1 | 99.1 | 1.1 | 99.1 | 1.1 |
| | IC-INVASE ($\lambda \uparrow 0.3$) | 96.8 | 1.0 | **96.4** | **0.4** | **96.4** | **0.4** | **96.4** | **0.4** |
| **Syn6** | INVASE (REP.) | 90.1 | 7.4 | | | | | | |
| | INVASE (EXP.) | 92.3 | 1.7 | 89.8 | 1.6 | 89.6 | 1.6 | 89.6 | 1.6 |
| | IC-INVASE ($\lambda \uparrow 0.2$) | 99.6 | 2.9 | 99.5 | 2.6 | 99.5 | 2.5 | 99.5 | 2.5 |
| | IC-INVASE ($\lambda \uparrow 0.3$) | 99.4 | 1.9 | **99.3** | **1.6** | **99.3** | **1.6** | **99.3** | **1.6** |

For each dataset, $20,000$ samples are generated and be separated into a training set and a testing set. In this work, we focus on finding outcome-relevant features (e.g., finding task-relevant actions in the context of RL), thus the true positive rate (TPR) and false discovery rate (FDR) are used as performance metrics.

**11-dim Feature Selection**    Table 2 shows the quantitative results of the proposed method, IC-INVASE on the 11-dim feature selection tasks. To accelerate training and facilitate the usage of dynamical computational graphs in curriculum learning and RL settings, the vanilla INVASE is re-implemented with PyTorch [23]. In general, the PyTorch implementation is 4 to 5 times faster than the previous Keras [1, 6] implementation, with on-par performance on the 11-dim feature selection tasks. In the comparison, both the reported results in  [38] (denoted by **INVASE (REP.)**) and our experimental results on INVASE (denoted by **INVASE (EXP.)**) are presented. The $p_r$ curriculum for IC-INVASE in all experiments are set to decrease from $0.5$ to $0.0$ except in ablation studies. Results of two different choices of the $\lambda$ curriculum are reported and denoted by **IC-INVASE** ($\lambda \uparrow \cdot$), e.g., $\lambda \uparrow 0.3$ means $\lambda$ increases from $0.0$ to $0.3$ in the experiment. We omit the results on the first three datasets (**Syn1,Syn2,Syn3**) where both IC-INVASE and INVASE achieve $100.0$ TPR and $0.0$ FDR. Iteration 1 to Iteration 4 in the table shows the results after applying the selection operator for different number of iterations.

In all experiments, IC-INVASE achieves better performance (i.e., larger TPR and lower FDR) than the vanilla INVASE with Keras and PyTorch implementation. Iterative applying the feature selection operator can reduce the FDR with a slight cost of TPR decay.

**100-dim Feature Selection**    We then increase the total number of feature dimensions to $100$ to demonstrate how IC-INVASE improves the vanilla INVASE in larege-scale variable selection settings. In this experiment. The features are generated with 100-dim Gaussian without correlations and the rules for label generation are still the same as the 11-dim settings. (i.e., 89 additional label-independent noisy dimensions of input is concatenated to the 11-dim inputs.)

The results are shown in Table 3. IC-INVASE achieves much better performance in all datasets, i.e., higher TPR and lower FDR. The ablation studies on different curriculum show both an increasing $\lambda$ and a decreasing $p_r$ can benefit discovery of label-dependent features. As the hyper-parameters for curriculum are not elaborated in our experiments, direct combining the two curriculum may hinder the performance. The design for curriculum fusion is left to the future work.

## C    Environment Details

**FourRewardMaze**    The FourRewardMaze is a 2-D navigation task where an agent need to find all four solutions to achieve better performance. The state space is 2-D continuous vector indicating the position of the agent, while the action space is a 2-D continuous value indicating the direction and step length of the agent, which is limited to $[-1, 1]$. The initial location of the agent is randomly

Table 3: Relevant feature discovery results for Synthetic datasets with 100-dim input

| DATA SET | METHOD | ITERATION 1 | | ITERATION 2 | | ITERATION 3 | | ITERATION 4 | |
|---|---|---|---|---|---|---|---|---|---|
| METRIC | | TPR | FDR | TPR | FDR | TPR | FDR | TPR | FDR |
| **Syn4** | INVASE (REP.) | 66.3 | 40.5 | | | | | | |
| | INVASE (EXP.) | 27.0 | 6.5 | 18.0 | 6.4 | 18.0 | 6.4 | 18.0 | 6.4 |
| | IC-INVASE W/O $p_r \downarrow$ | 66.3 | 40.5 | 66.3 | 40.5 | 66.3 | 40.5 | 66.3 | 40.5 |
| | IC-INVASE W/O $\lambda \uparrow$ | 100.0 | 43.0 | 100.0 | 43.0 | 100.0 | 43.0 | 100.0 | 43.0 |
| | IC-INVASE | **100.0** | 43.0 | **100.0** | 43.0 | **100.0** | 43.0 | **100.0** | 43.0 |
| **Syn5** | INVASE (REP.) | 73.2 | 23.7 | | | | | | |
| | INVASE (EXP.) | 56.4 | 37.9 | 56.4 | 37.9 | 56.4 | 37.9 | 56.4 | 37.9 |
| | IC-INVASE W/O $p_r \downarrow$ | 90.9 | 7.8 | 88.8 | 4.4 | 88.8 | 4.3 | 88.8 | 4.3 |
| | IC-INVASE W/O $\lambda \uparrow$ | 96.1 | 11.3 | 95.2 | 8.2 | **95.5** | 8.1 | **95.5** | 8.1 |
| | IC-INVASE | 91.9 | 8.1 | 90.8 | 4.3 | 90.8 | **4.2** | 90.8 | **4.2** |
| **Syn6** | INVASE (REP.) | 90.5 | 15.4 | | | | | | |
| | INVASE (EXP.) | 90.1 | 43.7 | 90.1 | 43.7 | 90.1 | 43.7 | 90.1 | 43.7 |
| | IC-INVASE W/O $p_r \downarrow$ | 98.5 | 4.1 | 98.4 | 2.4 | 98.4 | **2.3** | 98.4 | **2.3** |
| | IC-INVASE W/O $\lambda \uparrow$ | 99.6 | 8.1 | 99.6 | 7.1 | **99.6** | 7.0 | **99.6** | 7.0 |
| | IC-INVASE | 98.9 | 7.0 | 98.9 | 5.0 | **98.9** | 4.9 | **98.9** | 4.9 |

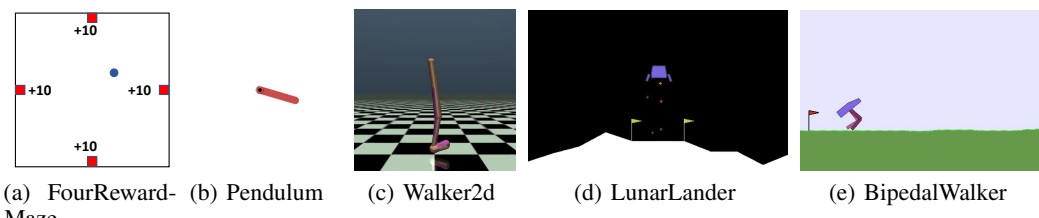

(a) FourReward-Maze    (b) Pendulum    (c) Walker2d    (d) LunarLander    (e) BipedalWalker

Figure 5: Environments used in experiments

selected for each game, and each episode has the length of 32, which is the timesteps needed to collect all four rewards from any starting position.

**Pendulum-v0** The Pendulum-v0 environment is a classic problem in the control literature. In the Pendulum-v0 of OpenAI Gym. The task has 3-D state space and 1-D action space. In every episode the pendulum starts in a random position, and the learning objective is to swing the pendulum up and keep it staying upright.

**Walker2d-v2** The Walker2d-v2 environment is a locomotion task where the learning objective is to make a two-dimensional bipedal robot walk forward as fast as possible. The task has 17-D state space and 6-D action space.

**LunarLanderContinuous-v2** In the tasks of LunarLanderContinuous-v2, the agent is asked to control a lander to move from the top of the screen to a landing pad located at coordinate $(0, 0)$. The fuel is infinite, so an agent can learn to fly and then land on its first attempt. The state is as 8-D real-valued vector and action is 2-D vector in the range of $[-1, 1]$, where the first dimension controls main engine, $[-1, 0]$ off, $[0., 1]$ throttle from $50\%$ to $100\%$ power and the second value in $[-1, -0.5]$ will fire left engine, while a value in $[0.5, 1.0]$ fires right engine, otherwise the engine is off.

**BipedalWalker-v3** The BipedalWalker-v3 is a locomotion task where the state space is 24-D and the action space is 4-D. The agent needs to walk as far as possible in each episode where a total timestep of 1000 are given and total 300 points might be collected up to the far end. If the robot falls, it gets $-100$ points. Applying motor torque costs a small amount of points, more optimal agent will get better score.

## D  Reproduction Checklist

### D.1  Neural Network Structure

In all experiments, we use the same neural network structure: in TD3, we follow the vanilla implementation to use 3-layer fully connected neural networks where $256$ hidden units are used. In the selector networks of the INVASE module, we follow the vanilla implementation to use 3-layer fully connected neural networks where $100, 200$ hidden units are used.

### D.2  Hyper-Parameters

In both TD-SWAR and the Dyn-SWAR, we apply IC-INVASE with $p_r$ reducing from $0.5$ to $0.0$ and $\lambda$ increasing from $0.0$ to $0.2$. While our experiments have already shown the effectiveness and robustness of those hyper-parameters, performing grid search on those hyper-parameters may lead to further performance improvement.

### D.3  Code

Our code is released anonymously at

https://anonymous.4open.science/r/Causal-RL-2718/.