# OpenReview forum: "Toward Causal-Aware RL: State-Wise Action-Refined Temporal Difference"
_NeurIPS.cc/2023/Conference — Submitted to NeurIPS 2023_

### Official Review · Reviewer_oAvw · 2023-07-04

**Soundness:** 2 fair
**Presentation:** 2 fair
**Contribution:** 2 fair
**Rating:** 5
**Confidence:** 3

**Summary:**

In this paper, the authors introduce a method called State-Wise Action Refined to explore the causal relationship between actions and task rewards in reinforcement learning. They address the issue of action space redundancy and propose interventions on the primal action space as a means to discover causality. The authors present two practical algorithms, TD-SWAR and Dyn-SWAR, which respectively identify task-related actions during temporal difference learning and uncover important actions through dynamic model prediction. These approaches not only provide insights into the decision-making process of RL agents but also enhance learning efficiency in tasks with redundant actions.

**Strengths:**



The authors proposed a method called State-Wise Action Refined to explore the causal relationship between actions and task rewards in reinforcement learning, which is significant and interesting.

**Weaknesses:**

The technical description of the proposed approach is complicated, and the main concern lies in the fact that the main technical contribution is not clear.

The paper contains some grammar mistakes and lacks sentence flow, making it difficult to understand the main contribution of the paper. For example, lines 34-36.

The proposed two practical algorithms, TD-SWAR and Dyn-SWAR, do not have sufficient experimental analysis support.

**Questions:**

See weaknesses

**Limitations:**

The technical description of the proposed approach is intricate, and the primary concern is the lack of clarity regarding its main technical contribution. The paper contains grammar mistakes and lacks sentence flow, making it difficult to understand the central contribution. For instance, lines 34-36 are particularly confusing. Additionally, the practical algorithms proposed, TD-SWAR and Dyn-SWAR, suffer from insufficient experimental analysis support,

---

> ### Author Rebuttal · Authors · 2023-08-07
>
> We sincerely appreciate the reviewer's efforts and constructive comments in improving our paper. Below please find our response:
>
> ---
> Our **technical contribution** is to introduce the state-wise action refinery based on the causal relationship in temporal difference learning to improve learning efficiency. Specifically, we present two practical algorithms, the TD-SWAR and the Dyn-SWAR. The former approach jointly learns the causal structure of the task and the policy, while the latter approach first constructs a causal relation through dynamics modeling and then performs TD learning.
>
> To better address the reviewer’s concern about the empirical evaluation, we’ve included
> 1. **More baselines** in the existing tasks.
> 2. **Incremental experiments** showing the sensitivity of the proposed method w.r.t. The degree of action redundancy.
> 3. **Additional discrete RL tasks** as benchmarks to provide more empirical evidence on the efficacy of the proposed methods.
>
> We would like to summarize our empirical studies and the conclusions as follows:
>
> - Empirical Study 1: Main Results with More Baselines
>
>   - **We use 100 redundant actions in the experiment to stress-test our proposed algorithms**. In those extremely challenging tasks, prevailing RL algorithms like TD3, SAC, and PPO all fail in efficient learning, yet our proposed method is able to conquer those challenges in most cases.
>
> - Empirical Study 2: Incremental Analysis on the Degree of Redundancy.
>
>   - We **additionally experiment with different numbers of redundant actions in the 5 environments**. The additional results are provided in the attached 1-page PDF (Figure 2). Comparing TD3, TD-SWAR, and Dyn-SWAR on the different settings, we find that TD3 is fragile to the action redundancy: in most cases, its performance drops sharply with an increasing number of redundancies. In comparison, **our proposed methods are more robust to the action space redundancy in general and are able to refine the action space and consistently improve the learning efficiency**.
>
> - Empirical Study 3: Extension to Discrete Action Space
>
>   - We provide **additional experiments** on the MiniGrid environment (discrete navigation) and provide a case study of our proposed method. In those tasks, action redundancy exists naturally (i.e., not all actions are useful for all tasks).
>
>   - To adapt the Dyn-SWAR algorithm to the discrete tasks, we combine it with DQNs (and call it Dyn-SWAR-DQN). Dyn-SWAR-DQN first learn the underlying causal-aware structure through dynamics modeling, and then when state-action values of DQNs are used in selecting the action, the arg-max operator only chooses actions within the selected subset.
>
>   - In our benchmark tasks, the agent needs to navigate to a goal position with partially observable visual inputs. The goal position may be located at a far position that requires the agent to pass through two different rooms/ gates before seeing it, hence introducing great challenge to learning algorithms.
>
>    - Performing exploration with such an action space will be much harder than exploring only the effective dimensions. In our experiments, we find Dyn-SWAR-DQN smartly **discovers the action dimensions causally related to the dynamical changes**, hence it only focuses on optimizing the Q-value estimation and performing exploration with such a subset. We can observe from the results (provided in the 1-page attached PDF file) that **such a refinery significantly improves the learning efficiency over the vanilla DQN**.
>
>
> - Empirical Study 4: Proof-of-Concept in Appendix.
>
>   - In pursuance of efficient learning through state-wise action refinery, a well-performing algorithm that is able to identify the actions that have a causal relationship with the given task is important. In order to scale up the instance-wise variable selection ability of INVASE, we introduce the two improvements over the existing algorithm and conduct ablation studies as well as stress tests in Appendix B.
>
>   - In this stress test, we increase the complexity of benchmarking tasks, and observe INVASE is not capable of solving high-dimensional input problems: its true positive discovery rate (TPR) decreases to 66%, 73%, and 90% in three challenging tasks (cf. Table 3. In Appendix B). On those tasks, our proposed method achieves TPR of 100%, 95.5%, and 99.6%, respectively. Showing the scalability of the improved instance-wise variable selection algorithm, which we used in identifying the causal-related actions in temporal difference learning.
>
> ---
> We’ve updated our manuscript accordingly to address your comments more explicitly.
> If there are leftover concerns, please let us know and we will do our utmost to address them.

---

> ### Author Response · Authors · 2023-08-16
> **A Gentle Reminder**
>
> It's been over a week since we shared our response addressing each point from the prior review comments, and we haven't yet had the privilege of receiving your feedback. To ensure we maximize the discussion period, we gently reach out, hoping the reviewer might engage further. We would appreciate it a lot if the reviewer could please let us know if there are any additional concerns or areas that need our clarification.
>
> In order to assist the reviewer in recalling the specifics of the work, the previous comments, and our response, we would like to offer a brief summary of each aspect:
>
> ## Summary of Our Work
> In our work, we tackle the problem of action space redundancy in reinforcement learning. We introduced the key insight of a causality-driven action space refinery and proposed two practical algorithms. We stress-test the proposed algorithms in various continuous control tasks with a highly redundant action space.
>
> ## Summary of Comments and Our Responses
> In your previous comments, you mainly had the following questions, and we answered those questions through the previous response:
>
> ### 1. Technical Contributions
>
> We argued our technical contributions include
> 1. Methodologically, we improve the instance-wise feature selection method to work under large-scale problems as temporal difference learning (empirical ablation studies provided in Appendix B)
> 2. Practically, we introduce the frameworks that integrate causality-driven action space refinery into temporal difference learning. This is further instantiated with two algorithms, the TD-SWAR, and the Dyn-SWAR.
> 3. Empirically, we provide evidence from various experiments to study the importance of action space refineries in action redundant tasks.
>
>
>
> ### 2. Experimental Analysis
>
> We provided additional experimental results during the rebuttal period. Our current experiments thoroughly demonstrate the efficacy of the proposed method, including
> 1. (For Proof-of-Concept) Experiments on synthetic data, 3 datasets, $\times$ 2 different settings.
> 2. (Main results, with ablation studies) Experiments on the 5 continuous control tasks, $\times$ 4 different settings.
> 3. (Extension to discrete RL, and case study) Experiments on the 3 discrete control tasks.
> We also included more baselines as suggested and observed from the results that those action space redundancy tasks are generally challenging for prevailing RL algorithms including PPO, SAC, and TD3.
>
> To sum up, we have **11 environments and datasets** (3 synthetics, 5 in the main text, and 3 discrete RL during rebuttal) in our evaluation. We stress-tested the proposed algorithms and provide ablation studies on those environments and verified all claims we made in the paper.
> That said, if there are suggestions for experiments that might improve our understanding of our method, we would be happy to run them!
>
>
> ---
>
> We genuinely value your perspective and were wondering if there might be any outstanding questions or concerns we can assist with. We would make the most of the time in the discussion period to address them.

---

> > ### Comment · Reviewer_oAvw · 2023-08-18
> >
> > Thanks to the authors for providing the rebuttals to address my concerns.  Most of my concerns have been addressed in a satisfactory manner.

---

> > > ### Author Response · Authors · 2023-08-19
> > > **Dear Reviewer oAvw**
> > >
> > > Once again, we deeply appreciate your follow-up feedback, insightful comments, time, and efforts devoted to refining our paper.
> > >
> > > We would like to kindly inquire if the resolution of your concerns could be reflected in the score. Your kind consideration will be much appreciated!
> > >
> > > Best Regards,
> > >
> > > Authors,

---

### Official Review · Reviewer_zze8 · 2023-07-06

**Soundness:** 2 fair
**Presentation:** 3 good
**Contribution:** 2 fair
**Rating:** 5
**Confidence:** 3

**Summary:**

This paper presents a strategy for action space reduction, based on causality-inspired modeling. It employs a method of action space selection, using collected data to identify the specific dimensions of actions that are meaningful to the task, as well as to improve the efficiency of exploration in reinforcement learning. The empirical evaluation confirms the effectiveness of the proposed module. Overall, the method is straightforward and can be embedded into most existing model-free and model-based RL frameworks. However, there are some details of the method and specific applications of causality, along with theoretical guarantees, that need clarification from the authors. Therefore, in the initial review, I give it a borderline score. If the authors provide the corresponding discussion and explanations, I will increase the score in subsequent reviews.






**Strengths:**

[**About Motivation and General Idea**] Even though there have been related studies, research on action reduction is very meaningful for Reinforcement Learning (RL), as it can improve sample efficiency and potentially even learning efficiency. The proposed approach of integrating causality to identify which actions have a causal relationship with the task reward provides an interpretable action reduction method.

[**About presentation**] The overall presentation is clear and easy to follow. The pipeline figure (Fig. 1) helps the readers to understand the steps.

[**About experiments**] Although there are similar baseline methods for comparison, overall, the experimental results validate the effectiveness of the method proposed in the paper, and the experimental design is fairly comprehensive. I will provide some additional suggestions regarding the experiments in a later section.

**Weaknesses:**

Note: Most of the weaknesses listed below are more like questions, discussion points, or suggestions, rather than outright flaws. Any clarifications provided by the authors would be very welcome and appreciated. Additionally, I don't have any personal interest in the related papers I suggested for reference. Given the limited time for rebuttal, it is not necessary to fully supplement experiments for comparison with these papers. Some clarifications and discussions on these methods would be greatly appreciated.

[**Regarding Intervention**] The authors mentioned that the causal discovery here is based on intervention, but there is a distinction between action and intervention. The authors could consider providing a clearer definition of intervention in this context, how interventions are conducted, and whether the target of each intervention is known, etc.

[**Regarding the Causality**] Although the entire pipeline is well designed and its effectiveness can be verified empirically, more explanation is needed on how to connect it with causality. From the method, it's not clear how to use causal properties for modeling or how to directly learn the causal relationships among states, actions, and rewards. The authors could consider providing some clarifications, either regarding theoretical guarantees (on the causal level) or presenting learned causal graphs, to check whether they empirically match the causal relationships between actions and state rewards in the environment. Here are some references: identifiability proof with action involved system: [1-2], and causal RL with learned causal graphs [3-4].



[**Aout related works**]

Regarding the discovery of causal RL and how to use causal structure to simplify the action space in RL, there are some related works. The authors could try to discuss them in the main text, possibly adding a subsection to introduce these works.

[**About Experiments**]

- For the experimental section, is it possible to visualize the learned sparse graph structure and compare it with the physical structure in real scenarios to see if they are similar?

- The current comparison does not involve related causal RL methods. It would be beneficial to discuss these, especially the one presented in [5], which uses a learned causal structure to find the minimal space—a very similar approach.

[**About Presentation**]

- In Figure 2, the graphical model is presented, but the title lacks some explanations, such as which dimensions have a causal relationship with the dynamics and reward, and what the blue and pink colors represent.

-  On line 195, does "figure 5" refer to "figure 3"? I noticed that figure 5 is in the appendix, and its content is similar to that of figure 3.

- In my opinion, it would be beneficial to first introduce the overall framework of action reduction and causal discovery, then provide specifics about its implementation within Reinforcement Learning, in conjunction with both model-free and model-based RL. This approach could also simplify algorithms 1 and 2, merging them into one algorithmic pipeline, detailing the implementations under different RL frameworks.

**References**

[1] Lachapelle, Sébastien, et al. "Disentanglement via mechanism sparsity regularization: A new principle for nonlinear ICA." Conference on Causal Learning and Reasoning. PMLR, 2022.

[2] Yao, Weiran, Guangyi Chen, and Kun Zhang. "Temporally disentangled representation learning." Advances in Neural Information Processing Systems 35 (2022): 26492-26503.

[3] Ding, Wenhao, et al. "Generalizing goal-conditioned reinforcement learning with variational causal reasoning." Advances in Neural Information Processing Systems 35 (2022): 26532-26548.

[4] Wang, Zizhao, et al. "Causal dynamics learning for task-independent state abstraction." arXiv preprint arXiv:2206.13452 (2022).

[5] Huang, Biwei, et al. "Action-sufficient state representation learning for control with structural constraints." International Conference on Machine Learning. PMLR, 2022.






**Questions:**

Most of the questions have been consolidated in the previous section, but there is still one remaining: for the optimization objective in eq.7, how can we theoretically and empirically ensure the learned structure is an identifiable causal structure? The authors could provide some insights, possibly in conjunction with some theoretical results from [1].

**References**

[1] Lachapelle, Sébastien, et al. "Disentanglement via mechanism sparsity regularization: A new principle for nonlinear ICA." Conference on Causal Learning and Reasoning. PMLR, 2022.

**Limitations:**

The authors briefly describe some limitations in the paper. I believe the main limitation might lie in how to theoretically link with the causality claimed in the article. From an empirical perspective, the work is relatively complete. Future work could consider making the entire framework scalable to more complex scenarios, such as MARL, etc.

---

> ### Author Rebuttal · Authors · 2023-08-07
>
> We deeply appreciate the reviewer's time and effort in evaluating our paper. We will now proceed to answer each of the posed questions:
>
> ---
> ### Q1: Intervention
>
> ### A1:
> In general, an _intervention_ in causal discovery is an external manipulation of a variable in the system, and we observe the effects on other variables to infer the causal relationship. In our context, we consider _interventions_ in a slightly extended sense, going beyond traditional definitions used in causal discovery.
>
> In our work, the _Selector Network_ plays an instrumental role in guiding the agent's interactions with the environment. While the RL agent's actions are a sort of _intervention_ on the environment that leads to state transitions, the **selector network essentially acts as a higher-level intervention mechanism**: it partially masks the output of the policy network, indirectly affecting the actions performed, consequently, the state transitions in the environment.
>
> From this perspective, we view the selector network as performing interventions on the policy network, altering its output. This is a form of indirect intervention where the target of intervention is the executed action, with the effects ultimately propagating to the environment.
>
> We acknowledge that this interpretation differs from typical interventions in causal discovery, where the target of intervention is a variable that directly influences outcomes. However, given the complex and dynamic nature of RL, we believe this extended view of 'intervention' is appropriate and insightful.
>
> In the revised manuscript, we have provided a detailed explanation of this extended concept of _intervention_ in our work, clarifying how the selector network influences the action and, subsequently, the state transitions in the environment. We hope this clarifies the use of _intervention_ in our work, and we are open to further discussions to enhance the clarity.
>
>
> ### Q2: Discovered Causal Structure as Case Study
>
> ### A2:
> We provide **additional experiments** on the MiniGrid environment (discrete navigation) and provide a case study of our proposed method. In those tasks, the action redundancy exists naturally (i.e., not all actions are useful for all tasks), hence we can verify the non-trivial causal structure discovered by our method.
>
> To adapt the Dyn-SWAR algorithm to the discrete tasks, we combine it with DQNs (and call it Dyn-SWAR-DQN). Dyn-SWAR-DQN first learns the underlying causal-aware structure through dynamics modeling, and then when state-action values of DQNs are used in selecting the action, the arg-max operator only chooses actions within the selected subset.
>
> In our benchmark tasks, the agent needs to navigate to a goal position with partially observable visual inputs. The goal position may be located at a far position that requires the agent to pass through two different rooms and gates before seeing it, hence introducing a great challenge to learning algorithms.
>
> Another challenge is its redundant action space, in the original design of the tasks, there are 7 actions:
>
>
> | Num | Name    | Action      |
> |-----|---------|-------------|
> | 0   | **left**    | Turn left   |
> | 1   | **right**   | Turn right  |
> | 2   | **forward** | Move forward|
> | 3   | pickup  | Unused      |
> | 4   | drop    | Unused      |
> | 5   | toggle  | Unused      |
> | 6   | done    | Unused      |
>
> _action space of the MiniGrid tasks, **bolded** denotes the action dimensions selected by Dyn-SWAR-DQN._
>
> Performing exploration with such an action space will be much harder than exploring only the effective dimensions. In our experiments, we find Dyn-SWAR-DQN smartly discovers that only the first three are **causally related** to the dynamical changes, hence it only focuses on optimizing the Q-value estimation and performing exploration with such a subset. We can observe from the results (provided in the 1-page attached PDF file, Figure 3) that **such a refinery significantly improves the learning efficiency over the vanilla DQN**.
>
>
> ### Q3: Complementary Discussions on Related Works
>
> ### A3:
>
> We provide **additional discussion on the mentioned related works**:
>
> GRADER [3] is specifically tailored for Goal-Conditioned RL tasks, utilizing the causal graph to enhance the generalization capabilities of RL algorithms. In contrast, our research emphasizes enhancing the efficiency of **single-task learning via action refinement**. The effectiveness of our method is **demonstrated across both continuous control tasks and discrete tasks**.
>
> Both CDL [4] and ASR [5] differ from our work as they put special emphasis on the **state space**. The work of CDL focus on improving the generalization ability of dynamics modeling model-based RL algorithms using causal awareness. The work of ASR focuses on state space representation learning. The work introduces a minimal set of state representations that suffice for policy learning. And the proposed SS-VAE works as a representation extractor for such a purpose. Differently, in our work, **we focus on the problem of action space redundancy**, and our proposed method works under the framework of TD learning — the model-free approach.
>
> Different from the identifiability theory literature [1-2] that has a general interest in causal discovery, our work put a specific focus on improving the learning efficiency of RL. In our context, the causal relationship between action, the state transitions, and corresponding returns are certain, yet the relationship between potentially redundant action dimensions and the outcomes is not known. And we proposed practical algorithms to perform causality-driven action refinery in our work.
>
> The discussion has been added to our related work section.
>
> ---
> We hope this response clears up your concerns, and we are more than willing for any further clarification and discussion.

---

> > ### Comment · Reviewer_zze8 · 2023-08-13
> >
> > I appreciate the authors for their thorough response.The majority of my concerns have been addressed satisfactorily.
> >
> > For better rigor regarding the concepts of causality and causal RL, I suggest that the authors more frequently refer to this approach as 'causality-based/driven RL'. This is due to the nuanced differences between the interventions described in this paper and those in causal discovery, as well as considerations of identifiability. While this isn't a strict requirement, a clear explanation, as promised by the authors, is essential.
> >
> > Once again, thank you for addressing my comments, and I look forward to following the ongoing discussion between the authors and other reviewers.

---

> > > ### Author Response · Authors · 2023-08-14
> > > **Thank You for the Encouraging Response**
> > >
> > >
> > > We sincerely appreciate the reviewer’s response and supportive feedback. We agree that using clearer descriptions is essential, especially when there is a potential risk of misunderstanding. We will continuously make our effort to enhance clarity. Below please find some revision examples in our updates:
> > >
> > > 1. In our abstract, we updated *In this work, we propose to conduct interventions on the primal action space to discover the causal relationship between the action space and the task reward.* to
> > > > In this work, we perform a causality-driven action space refinery, by focusing on only a subset of the action space that is causally related to the task reward.
> > >
> > > 2. In the introduction, we updated line 37 to
> > > >  With the proposed method, the agent learns to perform a special type of *intervention* with the selector network, which is designed for enhancing the causality awareness of RL. The selector network selects task-related actions according to the outcomes of its intervention over the policy networks...
> > >
> > > 3. In our method section, we explicitly updated the terminology to enhance clarity:
> > > > The objective of this work is to carry out a state-wise action refinery driven by the causal relationship between actions and decision outcomes. We introduce the selector network, which acts as a special type of intervention in the action space to discover the causal relationship during temporal difference learning.
> > >
> > > 4. In our updated Figure 1 caption, we explicitly explain how the selector network performs a special type of intervention on the action space:
> > > >  In our work, the selector network essentially acts as a higher-level intervention mechanism: it partially masks the output of the policy network, indirectly affecting the actions performed, consequently, the state transitions in the environment. From this perspective, we view the selector network as performing interventions on the policy network, altering its output. This is a form of indirect intervention where the target of intervention is the executed action, with the effects ultimately propagating to the environment.
> > >
> > > 5. In our method section, we consequently used the terminology of causality-driven action space refinery.
> > >
> > > ---
> > > We thank the reviewer again for the insightful comments, consideration, and help in improving our paper. We would like to kindly inquire if the resolution of your main concerns could be reflected in the score. We believe that the revisions and clarifications made with your help might align the manuscript more closely with higher criteria.

---

> > > > ### Comment · Reviewer_zze8 · 2023-08-15
> > > >
> > > > Many thanks for the detailed response. I believe that, with these changes, the clarity of this work will be enhanced. I am holding a positive review for this work and will be attentive to the ongoing discussion between the authors and other reviewers. I will raise my score if further discussions are also convincible.

---

> > > > > ### Author Response · Authors · 2023-08-21
> > > > > **Dear Reviewer zze8**
> > > > >
> > > > > We deeply appreciate the attention you have dedicated during our discussion period. As the current phase is approaching its conclusion, we would like to recap the enhancements made in light of feedback from our reviewers. We are keen to listen to your further advice, and we stand ready to provide any further clarifications if necessary. Your insights, perspective, and consideration are of paramount importance to our work. We genuinely value your expertise and would greatly appreciate your continued guidance on our work.
> > > > >
> > > > > During the discussion, we were honored that reviewers oAvw and dv5c also acknowledged and expressed satisfaction with our responses to their primary concerns. Moreover, reviewer dv5c updated the score from 5 to 6.
> > > > >
> > > > > We're also grateful for the constructive dialogue with reviewer 48ca, who highlighted:
> > > > > 1. The need for an in-depth delineation of the curriculum structure.
> > > > > 2. Potential extensions of our methods to different settings.
> > > > > 3. Clarification on our choice to employ causal language.
> > > > >
> > > > > To address those concerns and further refine our paper, we have additionally
> > > > >
> > > > > 1. provided **a thorough analysis and formal proof for the curriculum structure**.
> > > > > 2. provided additional empirical evidence in our supplemental experiments, including a series of ablation studies and new results on discrete environments, as suggested. And we explained how those results sufficiently **demonstrate the insights desired in our work.**
> > > > > 3. Incorporated **a specific section elucidating the motivation, rationale, and consideration of our presentation using causality.** (Please refer to the official comment "Remarks on Causality Interpretation" for more details).
> > > > >
> > > > > We have integrated all those updates into our manuscript. We're confident that by integrating their feedback, we have significantly enhanced the quality of our manuscript. We hope those updates could be helpful for your evaluation and consideration of our improvement.
> > > > >
> > > > > In the limited time remaining, we kindly request the reviewer to point out any areas of our work that may still need improvement. We are still eager to do our utmost to address them!
> > > > >
> > > > > Many thanks!
> > > > >
> > > > > Sincerely Regards,
> > > > >
> > > > > Authors

---

> > > > > > ### Comment · Reviewer_zze8 · 2023-08-21
> > > > > >
> > > > > > Thank you to the author for the further updates and feedback. I have read the discussion between the author and other reviewers, and I feel that the author has provided fairly reasonable explanations and responses to most of the issues raised. Overall, I think that some of the concerns raised by reviewer 48ca are valid, particularly regarding the use of causal language. However, I believe that the author can provide appropriate clarification and further refinement in the final version. Overall, I still maintain my positive rating. Once again, I appreciate the author's summary and heads-up

---

> > > > > > > ### Author Response · Authors · 2023-08-21
> > > > > > > **Dear Reviewer zze8**
> > > > > > >
> > > > > > > We sincerely appreciate your kind consideration and response!
> > > > > > >
> > > > > > > Yes, we do agree that the discussion with reviewer 48ca is insightful and helpful in refining our paper --- especially on the causality-related discussions. With the newly added section on causality interpretation, we believe the clarity of our presentation is further enhanced.
> > > > > > >
> > > > > > > Once again, we thank you for your time, effort, and consideration devoted to reviewing our paper!
> > > > > > >
> > > > > > > Regards,
> > > > > > >
> > > > > > > Authors

---

### Official Review · Reviewer_dv5c · 2023-07-07

**Soundness:** 3 good
**Presentation:** 3 good
**Contribution:** 3 good
**Rating:** 6
**Confidence:** 3

**Summary:**

The paper proposes an intervention-based method for understanding causality in Reinforcement Learning, aiming to enhance learning efficiency in tasks with redundant actions. The authors present two novel algorithms, Temporal Difference State-Wise Action Refined (TD-SWAR) and Dynamic State-Wise Action Refined (Dyn-SWAR). TD-SWAR identifies task-related actions during temporal difference learning, while Dyn-SWAR uncovers important actions through dynamic model prediction. The paper expands upon the recent advancements in instance-wise feature selection technology (INVASE), demonstrating how these improvements allow for more effective selection of task-related actions. Experiments underscore the effectiveness of causality-aware RL agents in action-redundant settings.

**Strengths:**

1. The paper tackles the underexplored topic of causality-awareness in RL, utilizing interventions on the action space to ascertain task-related actions. cal underpinnings.
2. The authors have made a clear effort to explain their work, providing intuitive examples such as the in-hand manipulation tasks and human learning analogies. The logical flow of the paper seems coherent.
3. The paper tackles a relevant issue in RL - action space redundancy and the importance of understanding causal relationships for learning efficiency. The findings have the potential to influence future RL research, particularly on exploration strategies and action space selection.

**Weaknesses:**

1. The action space of experiments environments are all very small. Given the known scaling limitation of causal discovery, it's important to discuss the capability of the proposed model as most of the real world RL environments are complex.
2. The paper could benefit from more comprehensive evaluation. They only compared with TD3 while there are many more advanced RL algorithms that have better performance on the experimented environments. Therefore it's not convincing that pruning action space is beneficial even in the simple environment.
3. The connection between the proposed methods and real-world application scenarios seems somewhat abstract. The authors could better explain how their proposed methods can help solve practical problems.
4. It's better to provide code to enhance reproducibility.
5. Formatting issue in line 6.

**Questions:**

1. It would be helpful to compare with state-of-the-art RL algorithms.
2. Could you please provide a discussion on the problem complexity (e.g., the number of variables) that the proposed method can handle? And discuss any limitations or potential pitfalls in their methods? This could include discussion on how robust their methods are to different types of environments or tasks. It's important to know whether these methods could be generalized across various RL applications.
3. It would be helpful if the authors could clarify how the proposed methods could be applied to real-world problems, and possibly provide case studies or scenarios demonstrating this.

**Limitations:**

The authors didn't address limitations nor broader societal impact, but I didn't see any ethical concerns.

---

> ### Author Rebuttal · Authors · 2023-08-07
>
> We thank the reviewer for the insightful comments and constructive feedback. We will respond to each of the questions in turn:
>
> ---
> ### Q1: More Comparisons, Baselines
>
> ### A1:
>
> In order to perform a more comprehensive evaluation and provide more empirical evidence, we include the **SAC and PPO as additional baselines** in our comparisons. We provide results in the attached 1-page PDF file (Figure 1). We observe both PPO and SAC fail in efficiently solving tasks with redundant action space.
>
> As a consequence, **the benchmark tasks are indeed hard enough for conventional RL algorithms** because the redundancy of action space introduces great difficulty to those algorithms. In our context, the _difficulty of environment_ could be manifested by the failure of existing RL algorithms. In comparison, refining the action space with our proposed algorithms can improve the learning efficiency significantly in those challenging tasks.
>
> We would provide additional experiment results by **changing the degree of redundancy**, as those results are also related to the second question on complexity analysis, please let us describe it with details in the response below.
>
> ### Q2: Discussion on the problem complexity, limitation, and stress tests.
>
> ### A2:
> We will answer this question from the following perspectives:
>   1. Complexity w.r.t. the degree of action redundancy (**with new experiments**)
>   2. Complexity w.r.t. the of dimension of states in continuous control (**with updated results**)
>
>
> - Complexity w.r.t. the degree of action redundancy
>
>   Although at first glance, it may seem unrealistic to use a large redundant action space, in fact, **we use 100 redundant actions in the experiment to stress-test different algorithms**.
>   In those extremely challenging tasks, existing RL algorithms like TD3, SAC, and PPO all fail in efficient learning, yet our proposed method is able to conquer those challenges in most cases.
>
>   To show such a stress test more explicitly, we **additionally experiment with different numbers of redundant actions in the 5 environments**. The additional results are provided in the attached 1-page PDF (Figure 2). Our proposed methods are more robust to the action space redundancy in general and are able to refine the action space and consistently improve the learning efficiency, compared to the baselines.
>
> - Complexity w.r.t. the number of states in continuous control
>
>   In our selected tasks, the state dimensions vary from 3 to 24, we find the 24-dim stochastic task BipedalWalker is **complicated enough** for the emergence of a performance pitfall — TD-SWAR is affected by such high environment stochasticity and Dyn-SWAR also failed to achieve the oracle performance (which is for sure a high standard).
> In the 17-dim Walker2d task, both TD-SWAR and Dyn-SWAR are able to achieve on-par performance with the oracle with 25 or 50 dimensions of action redundancy, yet another pitfall exists with 100 redundant actions being used in such a task.
>
> To sum up, **we stress-test the proposed algorithms by increasing the action redundancy, the state space complexity, and dynamics stochasticity**. We find Dyn-SWAR is more robust to all those variants, and the stability of TD-SWAR in highly stochastic large state space tasks can be further improved in future work (as we have discussed in the conclusion section).
>
>
> We’ve included this paragraph of discussion in the updated manuscript.
>
>
> ### Q3: Practical Problem and Case Study.
>
> ### A3:
> We provide **additional experiments** on the MiniGrid environment (discrete navigation) and provide a case study of our proposed method. In those tasks, action redundancy naturally exists as not all actions are useful for all tasks.
>
> In our benchmark tasks, the agent needs to navigate to a goal position with partially observable visual inputs. The goal position may be located at a far position that requires the agent to pass through two different rooms and gates before seeing the goal, hence introducing a great challenge to learning algorithms. We experiment with tasks with different levels of difficulty.
>
> To adapt the Dyn-SWAR algorithm to the discrete tasks, we combine it with DQNs and call it Dyn-SWAR-DQN. Dyn-SWAR-DQN first learns the underlying causal-aware structure through dynamics modeling and then uses the learned structure to perform action selection.
>
> In the original design of the tasks, there are 7 actions:
>
>
> | Num | Name    | Action      |
> |-----|---------|-------------|
> | **0**   | **left**    | Turn left   |
> | **1**   | **right**   | Turn right  |
> | **2**   | **forward** | Move forward|
> | 3   | pickup  | Unused      |
> | 4   | drop    | Unused      |
> | 5   | toggle  | Unused      |
> | 6   | done    | Unused      |
>
> _Table 1: action space of the MiniGrid tasks, **bolded** denotes the action dimensions selected by Dyn-SWAR-DQN._
>
> Performing exploration with such an action space will be much harder than exploring only the effective dimensions. In our experiments, we find **Dyn-SWAR-DQN smartly discovers that only the first three are causally related to the dynamical changes**, hence it only focuses on optimizing the Q-value estimation and performing exploration with such a subset. We can observe from the results (provided in the 1-page attached PDF file, Figure 3) that such a **refinery significantly improves the learning efficiency over the vanilla DQN**.
>
>
> ### Q4: Code for reproducibility.
>
> ### A4:
>
> We acknowledge the importance of reproducibility. Our code is provided in the anonymous link in Appendix D.3. Code for our additional experiments during the discussion period will also be released.
>
>
> ---
>
>
> We hope these clarifications address your concerns and further illuminate our ideas. Should there be any additional questions or concerns, we are more than willing to provide further explanations.

---

> > ### Author Response · Authors · 2023-08-16
> > **A Gental Reminder**
> >
> > It's been over a week since we shared our response addressing each point from the prior review comments, and we haven't yet had the privilege of receiving your feedback. To ensure we maximize the discussion period, we gently reach out, hoping the reviewer might engage further. We would appreciate it a lot if the reviewer could please let us know if there are any additional concerns or areas that need our clarification.
> >
> > In order to assist the reviewer in recalling the specifics of the work, the previous comments, and our response, we would like to offer a brief summary of each aspect:
> >
> > ## Summary of Our Work
> >
> > In our work, we tackle the problem of action space redundancy in reinforcement learning. We introduced the key insight of a causality-driven action space refinery and proposed two practical algorithms. We stress-test the proposed algorithms in various continuous control tasks with a highly redundant action space.
> >
> > ## Summary of Comments and Our Responses
> >
> > ### More Comparisons and Baseline
> >
> > ### Response
> > Following the reviewer's suggestion, we have additionally implemented PPO and SAC as baselines.
> >
> > ### Discussion on Complexity
> >
> > ### Response
> > We provided additional discussions on the complexity with regard to 1. degree of action space redundance and 2. degree of difficulty in continuous control.
> >
> > In fact, we have a section in our appendix that could also be helpful in addressing the reviewer's question on complexity: in Appendix B, we experimented with different numbers of redundancy on synthetic datasets to benchmark the performance of our proposed iterative curriculum learning paradigm.
> >
> >
> > ### Real-world Problems
> >
> > ### Response
> > We additionally experimented on existing environments that naturally contain action space redundancy, and showcase that our proposed method is able to identify the causally-related action dimensions out of the naturally redundant action space --- even in a discrete control setting. We also provide a case study to highlight how performance improvement is achieved. We would like to note that such improvement verifies the theoretical insights that have been found in previous studies of deep q learning. [1]
> >
> > ### Code for Reproducibility
> >
> > ### Response
> > Our code can be downloaded through the anonymous link provided in the paper. We will also release our code for the experiments added during the rebuttal period.
> >
> > ---
> > We genuinely value your perspective and were wondering if there might be any outstanding questions or concerns we can assist with. We would make the most of the time in the discussion period to address them.
> >
> > ---
> > __*Reference*__
> >
> > [1] Fan, Jianqing, et al. "A theoretical analysis of deep Q-learning." Learning for dynamics and control. PMLR, 2020.

---

> > > ### Comment · Reviewer_dv5c · 2023-08-20
> > >
> > > I would like to thank the authors for their detailed response. Most of my concerns have been addressed, and I have updated my score.

---

> > > > ### Author Response · Authors · 2023-08-20
> > > > **Thank You!**
> > > >
> > > > We sincerely appreciate the reviewer's positive feedback on our work.
> > > > Once again, we thank the reviewer for the time and effort devoted to reviewing and refining our paper!

---

### Official Review · Reviewer_48ca · 2023-07-07

**Soundness:** 1 poor
**Presentation:** 2 fair
**Contribution:** 2 fair
**Rating:** 2
**Confidence:** 4

**Summary:**

This paper shows a Q-learning-based RL algorithm that jointly performs feature selection to detect action variables that are not relevant to dynamics or rewards. The feature variable selection algorithm adopts INVASE, and the experiment confirms the idea of five artificially created extended environments.

----

During the rebuttal phase, we found that there are several major technical flaws around the causality.
Reflecting on such issues, I adjusted the score to the criteria.

**Strengths:**

The strength of the proposed approach is a simple extension of Q-learning-like algorithms that jointly perform action variable selection that improves the overall learning performance when there is a large number of redundant actions.

* originality: The idea of combining feature selection with INVASE during TD learning is novel
* correctness: The proposed idea looks correct
* clarity: The algorithm was presented in detail
* significance: This method may work well when there are large number of redundant actions

**Weaknesses:**

Although the title and Figure 2 mention “causal” and SCM, this paper is not relevant to causal discovery, or SCM is not relevant in the context. It can be better said as a kind of “feature selection.”
I am not sure how realistic it is to assume the nuisance actions and inflate action space to 100 by adding redundant actions in the experiment.

* Originality: The idea is original, and it could be improved if it considers settings common in causal discovery
* correctness: I believe the overall framework is correct, and I don't see a particular weakness
* clarity: This paper is not relevant to causal/SCM, which confused me by giving the impression that it performs causal discovery while TD-learning.
* significance: The experiment setting is unrealistic, and it limits assessing the significance

**Questions:**

1 In Proposition 1, How to ensure that a larger set of actions always contains the smaller ones?
In the proof, it is by definition of the operator, but it needs more explanation.

2 Is it possible to extend the proposed work to discrete action space?

**Limitations:**

I think this paper is not relevant for discussing limitations.

---

> ### Author Rebuttal · Authors · 2023-08-07
>
> We sincerely appreciate the reviewer's thoughtful insights and constructive feedback. We will respond to each of the comments and questions in turn.
>
> ---
> ### Point 1: Additional Experiment Results with Different Numbers of Nuisance Actions
>
> ### Reply 1:
>
> **We use 100 redundant actions to stress-test our proposed algorithms** in those extremely challenging tasks, prevailing RL algorithms like TD3, SAC, and PPO all fail in efficient learning, yet our proposed method is able to conquer those challenges in most cases.
>
> **Experiments under less redundancy also demonstrate the efficacy of our method** To better address your concern on the number of redundant actions, we **additionally experiment with different numbers of redundant actions in the 5 environments**. The additional results are provided in the attached 1-page PDF (Figure 2). Our proposed methods are able to refine the action space and improve the baseline under all settings: while TD3 is fragile to the action redundancy.
>
> **Our method also works on discrete tasks** in improving learning efficiency when the action redundancy naturally exists. Please see Answer 4 below.
>
> ### Point 2: The Relation to Causal Discovery
>
> ### Reply 2:
> Our work indeed involves a process of selecting certain dimensions or aspects of the action space, akin to feature selection. However, our approach goes beyond merely selecting these dimensions based on their predictive or explanatory power. Instead, we aim to uncover the causal relationships between selected action dimensions and resulting state transitions, drawing upon inspirations from causal discovery and SCMs.
>
> To be specific, **the _selector network_ in our approach operates on the premise of discovering causal components in the action space**, i.e., the subset of actions that lead to significant changes in the environment's state. This is essentially a process of causal discovery within the action space, where the _selector network_ performs a special type of _intervention_ — we acknowledge such an intervention is different from the conventional definition of causal discovery, given the complex and dynamic nature of RL. In this light, we believe our work aligns with both feature selection and causal discovery.
>
> However, we understand that our initial presentation may not have made this entirely clear, leading to a perception of our work as being primarily about feature selection. In the revised manuscript, we have made this clear that our method identifies and models causal relationships during temporal difference learning, as opposed to selecting useful features.
>
>
> ### Q3: Proposition 1
>
> ### A3:
> With Proposition 1, we show that the discovery of the underlying $M$ variables out of $L$ variables has a curriculum structure: for any $N_1, N_2$ that $M \le N_1 < N_2 \le L $, we can define the _$N_1$ problem_ as finding $N_1$ variables containing the $M$ truly outcome-related variables
> And similarly define a _$N_2$ problem_ as finding $N_2$ variables containing the $M$ truly outcome-related variables.
>
> While both optimizers in the $N_1$ and $N_2$ problems need to contain the $M$ outcome-related variables to minimize the KL divergence, those $N_1$ and $N_2$ selected variables do not necessarily have a subset relationship.
>
> Without loss of generality, in our construction of Proposition 1, we group those solutions to those problems with a subset relationship to show the curriculum structure. To show that  _the $N_1$ problem is harder than the $N_2$ problem._
>
> To see this, we can regard Proposition 1 as a **practical construction method** to show there exists Comb(L-M, N_2-N_1) (Comb: combinatorial number) many times more solutions to the $N_2$ problem than to the $N_1$ problem. Such a quantitative relationship becomes evident when we align solutions of the $N_1$ problem with their $N_2$ counterparts, emphasizing their subset relationship.
>
>
>
> ### Q4: Extension to Discrete Action Space.
>
> ### A4:
> Theoretically, it has been shown in [1] that the value estimation error bound of DQN is proportional to the dimension of action space $|\mathcal{A}|$ (Theorem 4.4). **Our action-refinery algorithm may achieve a linear acceleration by factor $\frac{|\mathcal{A}|}{|\mathcal{A}_\mathrm{refined}|}$ when redundancy exists.**
>
> We provide **empirical results on the discrete RL tasks to verify the theoretical insights and highlight the general applicability of our proposed method**. Specifically, we use the MiniGrid environment where not all actions designed in the environment are necessary for all tasks —- action redundancy naturally exists.
>
> In our benchmark tasks, the agent needs to navigate to a goal position with partially observable visual inputs. The goal position may be located at a far position that requires the agent to pass through two different rooms and gates before seeing the goal, hence introducing a great challenge to learning algorithms. We experiment with tasks with different levels of difficulty.
>
> To adapt the Dyn-SWAR algorithm to the discrete tasks, we combine it with DQNs and call it Dyn-SWAR-DQN. Dyn-SWAR-DQN first learns the underlying causal-aware structure through dynamics modeling and then uses the learned structure to perform action selection.
>
> In our experiments, we find **Dyn-SWAR-DQN smartly discovers the action dimensions that are causally related to the dynamical changes**, hence it only focuses on optimizing the Q-value estimation and performing exploration on such a subset. We can observe from the results (provided in the 1-page attached PDF file, Figure 3) that such a refinery **significantly improves the learning efficiency over the vanilla DQN in all tasks**.
>
> ---
>
> We hope that these clarifications address your concerns, and we are happy to have further discussions would they remain unclear.
>
> ---
> **References**
>
>
> [1] *Fan, Jianqing, et al. "A theoretical analysis of deep Q-learning." Learning for dynamics and control. PMLR, 2020.*

---

> > ### Comment · Reviewer_48ca · 2023-08-18
> > **Thanks for your response!**
> >
> > Thanks very much for addressing my questions, and it helps me to have a better understanding of your work.
> >
> > First of all, the explanation of Proposition 1 was helpful, but I still cannot understand why selecting a larger N is necessarily easier in a monotonic sense.
> >
> > Regarding the experiment, adding SAC/PPO as additional baselines, I think it is good to clarify two groups.
> > The first group solves the problem without any redundant actions, and the other handles redundant actions; this will show the gap and the motivation for using this action selection approach.
> >
> > Regarding alternative causality, this raises more concern. If you are developing a new foundation of causality that goes beyond existing ones, then it needs to be stated clearly rather than by your interpretation which is not consistent with causality theory.
> > In Figure 2, Dyn and R would be representing Dynamics and Reward in the problem. However, they are not defined as variables in SCM, which is irrelevant to SCM. I don't think this method completely differs from the existing casual inference/learning approach, but it simply does not show how it connects to the theory.
> > This is my opinion. If this method cannot be stated in terms of causal effect estimation, I think it is still OK to use this method, but it doesn't have to be phrased in terms of SCM and causal inference/learning. It is an action selection method that is still fine without causal interpretation. Maybe it would be better to bring this action selector for conventional causal discovery settings (finding causal graphs with interventional data) to clarify what it means by saying causal-aware.

---

> > > ### Author Response · Authors · 2023-08-19
> > > **Thanks for the Follow-up Feedback!**
> > >
> > > We deeply appreciate the reviewer's further clarifications and feedback on our response, we would try to address each of the leftover concerns in turn:
> > >
> > > ---
> > >
> > > > 1. Why selecting a larger N is necessarily easier in a monotonic sense?
> > >
> > > This is because of the monotonic change in exploration difficulty of the selector network, which is optimized through policy gradient.
> > >
> > > When using a large N, the optimization objective for the selector network essentially becomes finding any N action dimensions that are relevant to the temporal difference learning. In practice, such a step is implemented through the masking mechanism of the selector network. And the selector network will tend to output masks with N 1's, and L-N 0's. It will be **much more likely for the policy network to exploratively put a mask 1 onto a relevant dimension when N is larger**, rather than smaller.
> > >
> > > ---
> > > > 2. Show the gap and the motivation for using this action selection approach with experiments.
> > >
> > > We agree with the reviewer on the importance of showing the gap/ inherent difficulty of the action-redundant tasks. However, we would argue **the _group 2 experiments_ are sufficient** to "show the gap and the motivation for using this action selection approach."
> > >
> > > To be specific, the main consideration in benchmarking the performance of conventional RL algorithms in action-redundant tasks is to have a clear understanding of the difficulty, of those tasks. In our supplementary experiments, we observed the **relative** failure of PPO, SAC, and TD3 in those tasks. The **relative** inefficiency of DQN in solving action-redundant tasks also highlights such difficulty: as we have illustrated from both the theoretical perspective and empirical perspective, in our previous response.
> > > All of those experiment results provide clear evidence that directly applying existing RL algorithms is not an ideal option, whereas applying an action space refinery significantly improves learning efficiency.
> > >
> > > The other group of experiments mentioned by the reviewer corresponds to our "TD3-oracle" baselines. In those baselines, we reveal the true action space to TD3. Although such a setting is, undeniably, **an "unfairly" high-standard baseline**, we observed in our experiment that our proposed methods are able to achieve on-par performance in some tasks.
> > >
> > > On the other hand, we believe that including more "Oracle" performance on the curve would detract from, rather than complement, the presentation of our main results. That being said, if there are suggestions for experiments that might improve our understanding of the proposed methods, we would be happy to run them!

---

> > > > ### Comment · Reviewer_48ca · 2023-08-19
> > > > **Thanks for your response - continuing from question 1**
> > > >
> > > > Thanks for your explanation. However, I cannot understand why it has to be monotonic. The explanation sounds probably so, but such a procedure is done through neural networks. This statement can be said that if M is 2 and N1 and N2 are 1000,000,000 and 1000,000,001 then N2 is easier than N1. Or if M is 1000,000,000 and N1 is 1000,000,001 and N2 is 1000,000,002 then N2 is easier than N1. When it solves an easier problem the selector neural network deterministically finds the common action set again.
> > > > Could you explain how that happens? Or do you have an empirical study that verifies the statement?
> > > > What is expected to see is that by varying N1 and N2, the action set must be in a subset relation in various settings applies to the selector network.

---

> > > > > ### Author Response · Authors · 2023-08-19
> > > > > **Further Response to Question 1**
> > > > >
> > > > > We thank the reviewer for further clarification. We would like to respond with both proof and empirical results.
> > > > >
> > > > > ### (1) Formal Reasoning
> > > > > In this problem, it is not necessary nor helpful to consider N1/ N2 to be some concrete numbers. We would provide reasoning in symbolic language.
> > > > >
> > > > > To begin with, we re-state the notations and the claim, together with our previous explanation:
> > > > >
> > > > > We use M to denote the true numbers of relevant action dimensions, and L to denote the total action dimensions. We consider the problem of **Selecting exact N dimensions out of the L examples, such that those N dimensions include the M-relevant action dimensions.**
> > > > > We argue that such a problem is easier when N increases.
> > > > > And we explained that this is due to the change in the difficulty of exploration.
> > > > >
> > > > > To make it more explicit, by "exploration" we mean "the selector network discovers a feasible solution in random sampling over its action space."
> > > > >
> > > > > To see this, when selecting N dimensions out of L, there are Comb(L, N) different choices. On the other hand, when selecting N-M dimensions out of L-M dimensions, there are Comb(L-M, N-M) choices --- those choices correspond to the number of feasible solutions to the problem.
> > > > >
> > > > > Therefore, the probability of successfully finding a feasible solution is Comb(L-M, N-M)/Comb(L, N), which is **monotonically increasing when N increases**. The probability of finding feasible solutions to the problems in exploration with a larger N is higher than with a smaller N. The proof is then completed.
> > > > >
> > > > > We hope this more formal presentation can be helpful, and we will add those contents to the updated manuscript to enhance clarity.
> > > > >
> > > > > ### (2) Empirical Evidence
> > > > > We provide the above additional empirical evidence to demonstrate the effectiveness of the above curriculum, in our appendix B. As a proof-of-concept experiment, we use datasets benchmarked in the INVASE paper.
> > > > >
> > > > > On the last three tasks (which are considered to be the most challenging ones), our proposed method with curriculum learning improves performance significantly. We observed in our experiments (cf. Table 3. In Appendix B):
> > > > > - INVASE is not capable of solving high-dimensional input problems: its true positive discovery rate (TPR) decreases to 66%, 73%, and 90% in three challenging tasks.
> > > > > - Our proposed method achieves TPR of **100%, 95.5%, and 99.6%**, respectively. Showing the **importance and effectiveness of introducing a curriculum.**

---

> > > > ### Comment · Reviewer_48ca · 2023-08-19
> > > > **Continuing from question 2**
> > > >
> > > > This is my opinion. SAC/PPO is introduced as a new comparison.
> > > > What this paper shows is that the action selection method shown in this paper improves RL agent learning.
> > > > The basic expectation would be in an ideal setting where we fix the environment with a proper set of actions,
> > > > both algorithms (with and without action selection) show the same performance regardless of the RL algorithms TD3/DDQN/SAC/PPO).
> > > > Maybe there could be a small variation, but it should not be much different.
> > > > This is an Oracle environment setup.
> > > >
> > > > The next one is adding non-relevant actions to the environment and showing the degradation of performance
> > > > without an action selection algorithm, while the action selection method maintains a similar performance to the Oracle setup or less degradation.
> > > >
> > > > This way, the impact of the proposed algorithm would be clearly demonstrated.

---

> > > > > ### Author Response · Authors · 2023-08-19
> > > > > **Further Response to Question 2**
> > > > >
> > > > > We thank the reviewer for further clarification.
> > > > >
> > > > > ### (1) Results on TD3
> > > > > **In our attached pdf file, those results are already provided in Figure 2.** That said, we recognize that the layout and illustration of those figures are not ideal, limited by the page limit. We will sort those figures with a better grouping strategy, as suggested by the reviewers, in our revision.
> > > > >
> > > > > Specifically, the first row of figures provides results on TD3 with different degrees of action space redundancy; and the following two rows of figures provide results on our method. Comparing those results horizontally, we can see
> > > > >
> > > > > > adding non-relevant actions to the environment and showing the degradation of performance without an action selection algorithm
> > > > >
> > > > > as desired.
> > > > >
> > > > > ### (2) Results on Other Algorithms
> > > > > The suggestion provided by the reviewer is interesting in the way that it benchmarks different algorithms' robustness under action redundancy. We deeply agree that investigating such a direction can be an important and promising future research idea, yet we would argue that investigation is **out of the scope of our work, and can not add supplemental insight to our current discoveries.**
> > > > >
> > > > > To disclose the challenges and potential difficulties in such a promising future direction, we would note that those algorithms are distinct in their on/off-policy learning paradigm, policy learning objective, and regularizations. **Although applying our key insight of actino-space-refinery can be straightforward, the implementation and tailored design are non-trivial works.**
> > > > >
> > > > >
> > > > > ### (3) DQN
> > > > >
> > > > > **In our supplementary results (pdf file), we have shown how to combine DQN with our key insight, in an existing environment.** And this is a natural setting where action space is redundant.

---

> > > ### Author Response · Authors · 2023-08-19
> > > **Follow-Up Response (Continued)**
> > >
> > > > 3. Regarding causality and presentation
> > >
> > > If the consensus among reviewers leans towards presenting our work without the terminology of causality, we are amenable to such modifications. As has been pointed out by the reviewer, **our method doesn't have to be interpreted in the language of SCM and causal inference**. The essence of our insights, methodologies, and experimental findings remains significant even without invoking causal language.
> > >
> > > That said, we would also like to highlight our rationale and motivations in phrasing our method within the causality (causal-driven) context:
> > > 1. An inherent causal relationship between actions and state transitions naturally exists. Given this condition, it would be logical to delve into the detailed interplay between varying action dimensions and the corresponding state transitions. In our illustrative Figure 2, we utilize SCMs to illustrate the potential granular causal relationships between action dimensions and either (1) transitions (relating to the 'next state' variable), or (2) rewards (relating to the 'reward' variable). These cases correspond to our two practical algorithms and were meant to serve as intuitive foundational examples for action space refinery.
> > > 2. Traditional causal discovery approaches, which often rely on statistical tests, may not fully account for the dynamic intricacies inherent to temporal difference learning. Issues like the potential violation of the i.i.d. samples assumption may arise. Additionally, while RL offers the possibility of interventions, executing these requires further environmental assumptions, such as re-settability. This can limit their application. Recognizing these challenges, our pioneering methods employ the _selector network_ to perform perturbations using a binary masking process during temporal difference learning. We've chosen to view such perturbation as a unique form of intervention, as explained in our former response.
> > > 3. By interpreting our methods as "causality-driven", as opposed to “causal discovery”, we would like to emphasize the relatively underexplored area and the large potential of integrating causality with temporal difference learning.
> > >
> > > In conclusion, our primary aim is clarity and the avoidance of potential ambiguity. We remain flexible with both presentation phrasing choices, and concur with the reviewer that the choice of language—causal or not—will not alter the essence of our work. We commit to maintaining consistent terminology in our revision, contingent on feedback from the reviewers' discussion.
> > >
> > > ---
> > > We would appreciate it if the reviewers kindly let us know if there were any further questions. In the limited time remaining, we are still eager to do our utmost to address them!

---

> > > > ### Comment · Reviewer_48ca · 2023-08-19
> > > > **Continuing from question 3**
> > > >
> > > > Thanks very much for your response.
> > > > In my opinion, changing the text wouldn't help much as it weakens the statement that this method is doing causal learning, where it essentially learns a causal diagram.
> > > > As the paper claims a "causality-driven" or "causal discovery" algorithm, a proper comparison could be made against "causal discovery" algorithms available in the literature.
> > > > Or if authors decide to move away from causality, then it is closer to dimensionality reduction RL, a proper comparison could be made against those algorithms.
> > > >
> > > > Unfortunately, I cannot see why this method relates to SCM and causal discovery.
> > > > If this algorithm improves the continuous control environment, there are a collection of continuous control environments commonly evaluated beyond selected ones, which might be good to add to show since improvements can only be judged by experiments.
> > > > Would it be possible to apply this method to discrete action space?
> > > >
> > > > I think this paper has a lot of potential to address many interesting issues.

---

> > > > > ### Author Response · Authors · 2023-08-19
> > > > > **Further Response to Question 3**
> > > > >
> > > > > We thank the reviewer for further clarification.
> > > > >
> > > > >
> > > > > ### 1. Causal Interpretation (More detailed discussions please refer to official comments to all reviewers on this topic)
> > > > > Unfortunately, we find those follow-up comments **contradict your previous comments**. We would also remind there might exist confusion on the reviewer's side when causal discovery, inference, and effect estimation are interchangeably used.
> > > > >
> > > > > In the previous comment, it was said
> > > > >
> > > > > > If you are developing a new foundation of causality that goes beyond existing ones...
> > > > >
> > > > > Response: As an RL work, focusing on drawing insight from causal discovery, and performing causality-aware action space refinery, our work is definitely **not developing a new foundation of causality**. Hence we suppose we are the "or", as depicted later by the reviewer ---
> > > > >
> > > > > > If this method cannot be stated in terms of causal effect estimation, I think it is still OK to use this method, but it doesn't have to be phrased in terms of SCM and causal inference/learning. It is an action selection method that is still fine without causal interpretation.
> > > > >
> > > > > Response: Our method is inspired by and related to **causal discovery** (as opposite to emphasizing effect estimation). We might understand it wrongly, yet we thought the reviewer means we could phrase our method without SCM, and simply call it an action selection method, doing action-space-refinery.
> > > > >
> > > > > We would understand the **causal inference** in the above comments might be a **typo** of the reviewer, as the problem studied in our paper is on **causal discovery**, rather than *inference*.
> > > > >
> > > > > Considering the earlier feedback, we would appreciate further clarification on the subsequent comments regarding
> > > > >
> > > > > >> Changing the text wouldn't help much as it weakens the statement that this method is doing causal learning.
> > > > >
> > > > > >> I cannot see why this method relates to SCM and causal discovery.
> > > > >
> > > > > **May we ask the reviewer to kindly confirm and let us know whether you think we should present our work without "causal discovery"?**
> > > > > As we responded earlier, we do have several considerations on the necessity and rationale of using a causality language.
> > > > >
> > > > > Regardless of the presentation choices, we believe there is a need to provide the causality interpretation in a more formal way. And we would like to provide a detailed discussion through **official comments to all reviewers**. We hope the detailed discussion addresses the reviewer's concern about the causality language.
> > > > >
> > > > > ### 2. Causal Discovery Baselines
> > > > >
> > > > > First and foremost, we would like to note our work **first studies the problem of action space refinery under a causal discovery perspective, and there is no existing work that has bridged the gap that can work as a baseline.**
> > > > >
> > > > > Moreover, please kindly let us reiterate, conventional causal discovery approaches that rely on statistical tests, may not fully account for the dynamic intricacies inherent to temporal difference learning. Issues like the potential violation of the i.i.d. samples assumption may arise. Additionally, while RL offers the possibility of interventions, **executing these require further environmental assumptions, such as re-settability. This can limit their application.** Recognizing these challenges, our pioneering methods employ the selector network to perform perturbations using a binary masking process during temporal difference learning. We've chosen to view such perturbation as a unique form of intervention, as explained in our former response.
> > > > >
> > > > > Although **those reasons have been provided in our previous response**, we believe it is worth emphasizing them in case they were missed.
> > > > >
> > > > > ### 3. Would it be possible to apply this method to discrete action space?
> > > > >
> > > > > Although **we have already answered the reviewer's question and provided experiments in our initial response, and mentioned our experiments done with DQN in each of our follow-up responses**, we believe it is worth emphasizing them in case they were missed.
> > > > >
> > > > > The answer is affirmative, and we have provided both theoretical insights and empirical evidence in our response. We kindly request the reviewer to check the _author's rebuttal. Answer 4._; _A Gentle Reminder. Point 3_; _Replying to Thanks for your response! Answer 2_
> > > > >
> > > > >
> > > > >
> > > > > ---
> > > > > We would appreciate it if the reviewers kindly provide further clarifications. In the limited time remaining, we are still eager to do our utmost to address them!

---

> ### Author Response · Authors · 2023-08-16
> **A Gentle Reminder**
>
> It's been over a week since we shared our response addressing each point from the prior review comments, and we haven't yet had the privilege of receiving your feedback. To ensure we maximize the discussion period, we gently reach out, hoping the reviewer might engage further. We would appreciate it a lot if the reviewer could please let us know if there are any additional concerns or areas that need our clarification.
>
> In order to assist the reviewer in recalling the specifics of the work, the previous comments, and our response, we would like to offer a brief summary of each aspect:
>
> ## Summary of Our Work
> In our work, we tackle the problem of action space redundancy in reinforcement learning. We introduced the key insight of a causality-driven action space refinery and proposed two practical algorithms. We stress-test the proposed algorithms in various continuous control tasks with a highly redundant action space.
>
> ## Summary of Comments and Our Responses
> In your previous comments, you mainly had the following questions, and we answered those questions through the previous response:
> ### Point 1 Experiment Setting
>
> ### Answer 1:
> We addressed this concern by
> 1. Explaining the motivation of our experiment settings for stress-testing the algorithms;
> 2. Providing additional experiments to demonstrate the ability of our proposed method is general in different settings;
> 3. We added a more realistic scenario where action redundancy naturally exists.
>
> ### Point 2 Proposition 1
>
> ### Answer 2:
> We explained why the **definition of the operator will not lead to any loss of generality** in the previous response. Should it remain unclear, please kindly let us know and we are happy to expand and explain more on that point.
>
> ### Point 3 Extension to Discrete Action Space.
>
> ### Answer 3:
> We appreciate your suggestions on extending our method to the domain of discrete control. We have conducted experiments and highlighted the capability of extending our method to those discrete RL tasks. In three discrete navigation tasks, **our method achieves significant performance improvement over baselines**, by performing a causality-driven action space refinery.
>
> We believe the impact of our work is greatly enhanced after following your suggestion on showing efficacy in discrete environments.
>
> ---
> We genuinely value your perspective and were wondering if there might be any outstanding questions or concerns we can assist with. We would make the most of the time in the discussion period to address them.

---

### Author Rebuttal · Authors · 2023-08-07

We extend our sincere gratitude to all reviewers for their insightful comments, valuable suggestions, time, and efforts in evaluating and improving our paper.

---
To address the concerns raised by reviewers, we would respond to each of their questions respectively. Below, as a general response, we aim to outline the **key revisions and additional experimentation conducted by far**:

#### **Supplementary Experimental Evaluation** [Please refer to the attached PDF]

1. We added SAC and PPO as additional baselines to highlight the difficulty of the benchmark tasks used in our paper.
2. We conducted additional experiments by varying the degree of redundancy in the benchmark tasks.
3. We extended the proposed method to the discrete RL tasks and experimented on the MiniGrid domain as well as a case study.

#### **Revised Manuscript for Clarity** (Include but not limited to)

1. We made efforts to better illustrate how our algorithms are related, yet different from conventional causal discovery.
2. We extended our related work discussion to include several suggested works working on state space representation learning and identifiability theory.
3. We explicitly linked the selector network with intervention and explained how it works in Figure 1.
4. We clarified that our experiment settings and the tasks used are motivated by performing a stress test. And we additionally provide enriched settings including new experiment domains in pursuance of a more thorough evaluation.


----

We hope our clarification and additional empirical studies could address the concerns raised by reviewers. Should there be any leftover questions, please let us know and we will make every effort to address them during the subsequent discussion period.

---

### Author Response · Authors · 2023-08-15
**Invitation for Continued Feedback**

Dear Reviewers,

We deeply appreciate the insights you've shared during the review process. Following our revisions and previous responses, we are genuinely curious if we have adequately addressed the concerns you raised.

Should there be any leftover questions, concerns, or areas you feel need more clarification, please do not hesitate to let us know. We greatly respect your insights and stand ready to make any additional refinements based on your feedback.

Best regards,

Authors

---

### Author Response · Authors · 2023-08-19
**Remarks on Causality Interpretation**

We realized during the discussion phase of the rebuttal that the clarity of motivation, consideration, and rationale in phrasing our work in a causality language can be further enhanced. We would like to provide an additional explanation in the following section

---
## Remarks on Causality

### 1. Our Motivation for Using the Language of Causality is Three-fold:

- An inherent causal relationship between actions and state transitions naturally exists. Given this condition, it would be logical to delve into the detailed interplay between varying action dimensions and the corresponding state transitions. In our illustrative Figure 2, we utilize SCMs to illustrate the potential granular causal relationships between action dimensions and either (1) transitions (relating to the 'next state' variable), or (2) rewards (relating to the 'reward' variable). These cases correspond to our two practical algorithms and were meant to serve as intuitive foundational examples for action space refineries.

- Traditional causal discovery approaches, which often rely on statistical tests, may not fully account for the dynamic intricacies inherent to temporal difference learning. Issues like the potential violation of the i.i.d. samples assumption may arise. Additionally, while RL offers the possibility of interventions, executing these requires further environmental assumptions, such as re-settability. This can limit their application. Recognizing these challenges, our pioneering methods employ the selector network to perform perturbations using a binary masking process during temporal difference learning. Such perturbation can be regarded as a unique form of intervention.

- By phrasing our methods using the language of causality, we would like to emphasize the relatively underexplored area and the large potential of integrating causality with temporal difference learning. At the same time, **we would use causality-driven, as opposed to causal-discovery**, in our presentation to highlight the nuance difference and disclose we are not performing an intervention with the conventional approaches.


### 2. Causal Discovery on Observational Decision Data in RL is Hard
Although the causal relationship between actions, states, and next states is certain, the fine-grained causal relationship regarding different dimensions of actions is unclear. And performing discovery on observational decision data is hard.

The difficulty in causal discovery from the decision dataset arises from the fact that those **actions are not randomly generated**, rather, they are sampled from the policy $\pi$. In such a case, it is, in general, impossible to perform causal discovery. Hence we are not able to perform action-space refinery according to a causal relationship just based on observational decision data. Intuitively, we may consider using interventions to address such an issue, yet there are several challenges that hinder the direct usage of conventional intervention methods.


### 3. Challenges in Applying Conventional Interventions

While the interactive dynamics of RL offer opportunities for causal discovery through interventions, the direct application of current intervention methods presents numerous challenges.

- 1. To perform an intervention, the environments **must be resettable** to any state. This is a strong assumption that can not be satisfied by **many** simulated tasks, and **can not be satisfied by any** real-world task.
- 2. The intervention should be performed to each dimension of the action space, making it intrinsically inefficient and infeasible when the action space is large.
- 3. The intervention practice for continuous actions is unclear, and defining proper interventions on each dimension of those continuous actions requires non-trivial effort.
- 4. It faces the curse of dimensionality when facing continuous state tasks, where enumerating the state space is impossible.


### 4. Our Method: A "Reparameterization Trick"

In our work, we introduce the selector network to act as a special type of intervention. The selector network plays an instrumental role in guiding the agent's interactions with the environment. While the actions are a sort of intervention on the environment that leads to state transitions, the selector network essentially acts as a higher-level intervention mechanism: it partially masks the output of the policy network, affecting the actions performed, consequently, the state transitions in the environment.

---

> ### Author Response · Authors · 2023-08-19
> **Remarks on Causality Interpretation (Continued)**
>
> Formally, we can present the DAGs of the two approaches: in DAG1, intervention in the action space is hard due to the reasons illustrated above. DAG2 illustrates how our work uses the selector network G to circumvent those difficulties:
>
>
> _DAG 1: The original problem of causal discovery requires intervention on **each of the action dimensions**, which is hard to randomize during learning_
> |     |  |  |     |     |
> |-----|-----|---|-----|-----|
> |     | $\pi$ |  |     |     |
> |     |  ↓   |  |     |     |
> |     | $a_i$   |   | $s$   |     |
> |     | ↘   |   | ↓   |     |
> |     |     |   |   $s'$ |     |
>
> _DAG 2: Our approach "reparameterizes" the problem by performing an **intervention with the selector network G**, the mask can be randomized during learning_
> |    |       |  |  |     |     |
> |-----|-----|-----|---|-----|-----|
> |  G |       | $\pi$ |  |     |     |
> |   ↓ |       |  ↓   |  |     |     |
> |   $m$  |     | $a$   |   | $s$  |       |
> |    |  ↘     | ↓   | ↙  |   |     |
> |    |       | $s'$   |    |    |     |
>
>
> Essentially, our approach can be analogized as a "parameterization trick", that performs interventions through masking rather than directly to the action.
>
> - Solving 1. With our Equation (6), the learning objective is to minimize the expected difference between applying and not applying G. Such property enables the treatment effect estimation computed on a population level, circumventing the difficulty of resetting environments.
> - Solving 2. It performs intervention in the mask space, which is discrete for both discrete and continuous control tasks. Importantly, different action dimensions are binarily encoded, such that we do not need to perform intervention for each action dimension individually anymore.
> - Solving 3. The definition of intervention in the mask space is clear, and binary.
> - Solving 4. We combine the objective of temporal difference learning with causal discovery through Equation (6). Optimizing such an objective simultaneously improves the performance of both.
>
> Equation (6), which is our learning objective of the selector network G, can be explicitly interpreted in DAG2 as selecting the dimension of actions that have a non-trivial treatment effect. From such a perspective, the selector network is performing a special type of intervention.
>
>
> We acknowledge that this interpretation differs from typical interventions in causal discovery, yet given the complex and dynamic nature of RL, we believe our approach addressed the issues of conventional intervention discussed above appropriately and insightfully --- using the language of causality.
>
> ---
>
> We hope our comments on causality have added clarity to our presentation. If there are any concerns, we genuinely ask the reviewers to inform us. We are committed to addressing them to the best of our ability!

---

### Author Response · Authors · 2023-08-21
**Dear Area Chairs,**

Firstly, we'd like to express our gratitude for your leadership in steering the review process. We recognize the tremendous responsibility and effort this entails.

During the discussion phase, reviewers zze8, oAvw, and dv5c showed a positive reception to our responses, with most of their concerns being addressed. We note that not all this acknowledgment is reflected in the scores yet, and we respect the reviewers' discernment in their evaluations. Collectively, our scores stand positively at 5, 5, and 6.

However, while we do value the feedback of reviewer 48ca, and recognize its role in refining our paper, there were moments where **the reviewer's perspective seemed somewhat skewed.** We faced challenges discerning the exact basis for certain comments, as they at times appeared contradictory.

The original rating of reviewer 48ca was expressly based on a set of criticisms/questions that have since been addressed in great detail:

1. Whether our method works in more “realistic” environments in terms of the number of redundant actions:
- **As response**, we provided **additional experiments as suggested by the reviewer,** demonstrated our methods work in all different settings, and **explained our original motivation of stress-testing.**
2. Explanation of proposition:
- **As response**, we provided a **formal proof**, as well as **experiment analysis.**
3. Experiments on discrete action space:
- **As response**, we provide **additional experiment results as suggested by the reviewer**, and **explicitly explained** those results in our initial response, the reviewer **asked exactly the same question again** in the last reply.
4. Causal Interpretation:
- **As response**, we added an **additional section “remarks on causal interpretation”**, to explicitly explain the motivation, rationale, and necessity of using causal language.


Despite our thorough responses, inclusive of new results, detailed proofs, and clarified content, we noted the reviewer keeps silent after our clarifications are posted. This raised concerns about whether our diligent efforts to rectify and elaborate on our work have been duly acknowledged.

**We would like to gently argue the reviewers' unfair comments could potentially compromise our fair chances of being considered for presentation at the conference, and wish this could be put in kind consideration during the following reviewer discussion period**, to be specific, in reviewer 48ca's comments

1. There were **inconsistencies** regarding the use of causal language.
2. Terms like causal discovery and causal inference are used interchangeably, **by mistake.**
3. Requests for additional experiments **often lacked solid motivation**, especially when such **results had already been showcased.** --- the repeated request for existing results.

We would appreciate it if the reviewers kindly let us know if there were any further questions. In the extremely limited time remaining, we are still eager to do our utmost to address them!

Many thanks!

Regards,

Authors

---

> ### Comment · Reviewer_48ca · 2023-08-21
> **Response to the response of authors**
>
> Authors pointed out that my response has some flaws and this is my explanation to such statements.
>
>
>  On proposition 1.
> The issue that I saw was that the optimization is done by fitting neural networks.
> So, there may exist several things that influence the output in non-deterministic manner,
> such as initialization, hyper-parameter selection, convergence of parameters.
> Considering such aspects, is proposition 1 still valid on any setting?
> The statement was that increasing N will maintain the monotonic property, the explanation was mentioning stochastic aspect too.
>
>
> On Figure 2 and terminologies in causality.
> Clearly Figure 2 is not an SCM for TD learning in RL.
> What authors may want to say would be some actions are not causally relevant to Dyn/R
> Independent to the type of the test employed for learning causal grpahs,
> it boils down toward estimating the causal effects with/without such variables.
> In TD-learning setting, computing the expected reward under some interventional distribution can be seen as performing causal inference.
> Note that there's no inconsistency in the usage of terms and it is not by mistake.
>
>
> On the author's statements on interventions.
> One difficulty that I see in this paper and the response is that
> the terminology is not consistent to causality literature and some statements are wrong.
>
> Let's take a look at "3 Challenges in Applying Conventional Interventions".
> To perform an intervention, the environments must be resettable to any state.
>
> this is not true.
> the essence of causal inference is being able to estimate causal effects when you cannot have luxury of having such an ideal environment.
>
> The intervention should be performed to each dimension of the action space, making it intrinsically inefficient and infeasible when the action space is large.
>
> this is not true.
> there's no reason that one should think of intervention applied to each dimension.
>
> The intervention practice for continuous actions is unclear, and defining proper interventions on each dimension of those continuous actions requires non-trivial effort.
> It faces the curse of dimensionality when facing continuous state tasks, where enumerating the state space is impossible.
>
> this is not true.
> causal effect in continuous space is well defined.
>
>
> On reparameterization trick part.
> What's the meaning of a special type of intervention?
> what makes something conventional intervention and sometihng special?
> What it means by syaing a higher-level intervention?
> is there a lower-level intervention?
> What it means by saying "instrumental role"?
> All those statements are informal handwavy that doesn't really say anything.
>
>
> On the experiment.
>  Authors added SAC/PPO to the Figure and claimed that the environment is challenging.
> I saw that it only confuses, and it is better to add oracle/non-oracle set up as done with TD3 if authors wanted to add SAC/PPO to the experiment. This implicitly assumes that the performance of SAC and TD3 will be similar in all environments, but we often see that it heavily depends on hypreparameter settings.
>
>
> From the response to other reviews, showing DAG1 and DAG2.
>
> It is misleading or wrong diagram.
> There is an arrow from state s to action a missing.
> For causal relevance, we test for outcome variables and s' is not an outcome variable.

---

> > ### Author Response · Authors · 2023-08-21
> >
> > The reviewer posted that response after the discussion deadline, we would do our best to address those comments given the extremely limited time, as the system opens.
> >
> > Respond to Q1. The reviewer's comments on neural networks are misleading. The curriculum nature is irrelevant to using NN approximators. The reviewer confuses what is a problem and what is a solver of a problem.

---

> > ### Author Response · Authors · 2023-08-21
> >
> > Response to Q2. The reviewer **is wrong in saying**
> >
> > > estimate causal effects when you cannot have luxury of having such an ideal environment.
> >
> > Discovering causal relationships from non-randomized observational data is **impossible**. This is the very basic concept well-acknowledged in causal discovery.

---

> > ### Author Response · Authors · 2023-08-21
> >
> > Response to Q3.
> >
> > Our initial response was released before 10 Aug. The reviewer only replied 4 days before the discussion period ends. It is unfair to ask for random experiments without solid rationale.
> >
> > Although it is clear in our work we focused on continuous control, the reviewer suggests us in adding experiments on the discrete tasks. We finished those experiments, yet the reviewer never acknowledge this. Instead, the **reviewer asks for more experiments without solid reasoning.**
> >
> > As we have explained, although the insight of transferring our insight into other RL algorithms is straightforward, PPO and SAC have their own properties, hence such adaptation is non-trivial. Studying how to transfer our idea to other algorithms does not add insight to this work.
> >
> > And this is especially unfair to be asked at the ending stage of the discussion.

---

> > ### Author Response · Authors · 2023-08-21
> >
> > That said. We would like to express our most sincere thanks to reviewer 48ca, for the criticism, time, and effort devoted to reviewing and improving our paper!
> >
> > Although we are sorry that on several points we can not reach an agreement and believe the reviewer holds somehow biased opinion, we would take those comments into consideration in our revision, make our presentation more clear and avoid misunderstanding for future readers.
> >
> > Once again, we thank our reviewers for their consideration and evaluation of our paper.

---

### Decision · Program_Chairs · 2023-09-21

**Decision:**

Reject

**Comment:**

The paper introduces a Q-learning RL algorithm that uses causality-inspired approach to identify irrelevant action variables via INVASE. New algorithms, TD-SWAR and Dyn-SWAR, enhance learning efficiency in tasks with redundant actions. This paper is considered to be novel (dv5c). However, It has concerns with technical flaws (dv5c) and clarity issues (oAvw,zze8). The decision is the rejection.